# DyGMamba: Efficiently Modeling Long-Term Temporal Dependency on Continuous-Time Dynamic Graphs with State Space Models

**Zifeng Ding**[*]                                                           *zd320@cam.ac.uk*
*University of Cambridge*

**Yifeng Li**[*]                                                              *yifeng.li@tum.de*
*Technical University of Munich*

**Yuan He**                                                                 *yuan.he@cs.ox.ac.uk*
*University of Oxford*

**Antonio Norelli**                                                   *antonio.norelli@cs.ox.ac.uk*
*University of Oxford*

**Jingcheng Wu**                                             *jingcheng.wu@ki.uni-stuttgart.de*
*University of Stuttgart*

**Volker Tresp**                                                       *tresp@dbs.ifi.lmu.de*
*LMU Munich*

**Michael Bronstein**[†]                                     *michael.bronstein@cs.ox.ac.uk*
*University of Oxford*

**Yunpu Ma**[†]                                                  *cognitive.yunpu@gmail.com*
*LMU Munich, Munich Center for Machine Learning*

**Reviewed on OpenReview:** *https://openreview.net/forum?id=sq5AJvVuha*

## Abstract

Learning useful representations for continuous-time dynamic graphs (CTDGs) is challenging, due to the concurrent need to span long node interaction histories and grasp nuanced temporal details. In particular, two problems emerge: (1) Encoding longer histories requires more computational resources, making it crucial for CTDG models to maintain low computational complexity to ensure efficiency; (2) Meanwhile, more powerful models are needed to identify and select the most critical temporal information within the extended context provided by longer histories. To address these problems, we propose a CTDG representation learning model named DyGMamba, originating from the popular Mamba state space model (SSM). DyGMamba first leverages a node-level SSM to encode the sequence of historical node interactions. Another time-level SSM is then employed to exploit the temporal patterns hidden in the historical graph, where its output is used to dynamically select the critical information from the interaction history. We validate DyGMamba experimentally on the dynamic link prediction task. The results show that our model achieves state-of-the-art in most cases. DyGMamba also maintains high efficiency in terms of computational resources, making it possible to capture long temporal dependencies with a limited computation budget[1].

---

[*]Equal Contribution. Zifeng's work done during the visit at the University of Oxford.
[†]Corresponding author.
[1]We release our code at the following link: https://github.com/ZifengDing/DyGMamba.

# 1 Introduction

Dynamic graphs store node interactions in the form of links labeled with timestamps (Kazemi et al., 2020). In recent years, learning dynamic graphs has gained increasing interest since it can be used to facilitate various real-world applications. Dynamic graphs can be classified into two types, i.e., discrete-time dynamic graph (DTDG) and continuous-time dynamic graph (CTDG). A DTDG is represented as a sequence of graph snapshots that are observed at regular time intervals, where all the edges in a snapshot are taken as existing simultaneously, while a CTDG consists of a stream of events where each of them is observed individually with its own timestamp. Previous works (Kazemi et al., 2020; Shirzadkhani et al., 2024) have indicated that CTDGs have an advantage over DTDGs in preserving temporal details, and therefore, more attention is paid to developing novel CTDG modeling approaches for dynamic graph representation learning.

Recent effort in CTDG modeling has resulted in a wide range of models. However, most of them are unable to model long-term temporal dependencies of nodes, despite the existence of abundant historical information. To solve this problem, Yu et al. (2023) propose a CTDG model DyGFormer that can handle long-term node interaction histories based on Transformer (Vaswani et al., 2017). Despite its ability in modeling longer histories, employing a Transformer naturally introduces excessive usage of computational resources due to its quadratic complexity. Another recent work CTAN (Gravina et al., 2024) tries to capture long-term temporal dependencies by propagating graph information in a non-dissipative way over time with a graph convolution-based model. Despite the model's high efficiency, Gravina et al. (2024) show that CTAN cannot capture very long histories and is surpassed by DyGFormer on the CTDGs where learning from very far away temporal information is critical. Based on these observations, we summarize the first challenge in CTDG modeling: **How to develop a model that is scalable in modeling very long-term historical interactions?** Another point worth noting is that as longer histories introduce more temporal information, more powerful models are needed to identify and select the most critical parts. This reveals another challenge: **How to effectively select critical temporal information with long node interaction histories?**

To address the first challenge, we propose to leverage a popular state space model (SSM), i.e., Mamba SSM (Gu & Dao, 2023) to encode the long sequence of historical node interactions. Since Mamba is proven effective and efficient in long sequence modeling (Gu & Dao, 2023), it maintains low computational complexity and is scalable in modeling long-term temporal dependencies. For the second challenge, we address it by learning temporal patterns of node interactions and dynamically selecting the critical temporal information based on them. The motivation can be explained by the following example. Consider a CTDG with nodes as people or songs and edges representing a person playing a song at a specific time. If a person $u$ frequently plays a hit song $v$ initially but decreases the frequency later on, the time intervals between plays increase. Ignoring this pattern can lead models to incorrectly predict that $u$ will still play $v$ at future timestamps due to their high appearances in each other's historical interactions. If a CTDG model recognizes this pattern, it can prioritize other temporal information, such as $u$ increasingly listening to a new song $v'$ before $t$, instead of focusing on $u$, $v$ interactions. Since each pattern corresponds to a specific edge, e.g., $(u, v, t)$, we name these patterns as edge-specific temporal patterns.

To this end, we propose a new CTDG model named DyGMamba. DyGMamba first leverages a node-level Mamba SSM to encode historical node interactions. Another time-level Mamba SSM is then employed to exploit the edge-specific temporal patterns, where its output is used to dynamically select the critical information from the interaction history. To summarize: (1) We present DyGMamba, the first model using SSMs for CTDG representation learning; (2) DyGMamba demonstrates high efficiency and strong effectiveness in modeling long-term temporal dependencies in CTDGs; (3) Experimental results show that DyGMamba achieves new state-of-the-art on dynamic link prediction over most common CTDG datasets.

# 2 Related Work and Preliminaries

## 2.1 Related Work

**Dynamic Graph Representation Learning.** Dynamic graph representation learning methods can be categorized into two groups, i.e., DTDG and CTDG methods. DTDG methods (Pareja et al., 2020; Goyal

et al., 2020; Sankar et al., 2020; You et al., 2022; Li et al., 2024a) can only model DTDGs where each of them is represented as a sequence of graph snapshots. Modeling a dynamic graph as graph snapshots requires time discretization and will inevitably cause information loss (Kazemi et al., 2020). To overcome this problem, recent works focus more on developing CTDG methods that treat a dynamic graph as a stream of events, where each event has its own unique timestamp. Some works (Trivedi et al., 2019; Chang et al., 2020) model CTDGs by using temporal point process. Another line of works (Xu et al., 2020; Ma et al., 2020; Wang et al., 2021b; Gravina et al., 2024) designs advanced temporal graph neural networks for CTDGs. Besides, some other methods are developed based on memory networks (Rossi et al., 2020; Liu et al., 2022), temporal random walk (Wang et al., 2021c; Jin et al., 2022) and temporal sequence modeling (Cong et al., 2023; Yu et al., 2023; Tian et al., 2024). Since some real-world CTDGs heavily rely on long-term temporal information for effective learning, a number of works start to develop CTDG models that can do long range propagation of information over time (Yu et al., 2023; Gravina et al., 2024).

**State Space Models.** Transformer (Vaswani et al., 2017) is a de facto backbone architecture in modern deep learning. However, its self-attention mechanism results in large space and time complexity, making it unsuitable for extremely long sequence modeling (Duman Keles et al., 2023). To address this, many works focus on building structured state space models that scale linearly or near-linearly with input sequence length (Gu et al., 2021; 2022b;a; Smith et al., 2023; Peng et al., 2023; Ma et al., 2023; Gu & Dao, 2023). Most structured SSMs exhibit linear time invariance (LTI), meaning their parameters are not input-dependent and fixed for all time-steps. Gu & Dao (2023) demonstrate that LTI prevents SSMs from effectively selecting relevant information from the input context, which is problematic for tasks requiring context-aware reasoning. To solve this issue, Gu & Dao (2023) proposes S6, also known as Mamba, which uses a selection mechanism to dynamically choose important information from input sequence elements. Selection mechanism involves learning functions that map input data to SSM's parameters, making Mamba both efficient and effective in modeling language, DNA sequences, and audio.

**State Space Models for Graphs.** Recently, there have been several works employing Mamba SSM for representation learning on static graphs (Wang et al., 2024; Behrouz & Hashemi, 2024). Wang et al. (2024) integrate a Mamba SSM into graph transformers, employing input-dependent node selection and prioritization mechanisms to efficiently capture long-range multi-hop dependencies in static graphs. Behrouz & Hashemi (2024) adapt Mamba by creating a bidirectional SSM for static graphs, utilizing hierarchical tokenization and an optional positional encoding mechanism to improve scalability and performance on diverse graph-based downstream tasks. Both of them aim to efficiently encode long-range multi-hop graph information. Unlike our work, they do not have dedicated structures to process temporal information and are thus not suitable for CTDG modeling. There is also another work (Li et al., 2024b) that tries to use Mamba to model spatial-temporal graphs (STGs). Li et al. (2024b) employ a Spatial-Temporal Selective State Space Module for adaptive spatial-temporal feature selection and Kalman Filtering Graph Neural Networks to dynamically integrate and optimize embeddings from various temporal granularities. Although this work achieves high efficiency and superior forecasting accuracy on STGs, it is not applicable to modeling CTDGs because (1) STGs are inherently DTDGs, and (2) this model primarily aims at predicting node states in STGs (e.g., traffic sensor readings) rather than explicitly capturing the continuous and irregular structural evolution characteristic of CTDGs. In our work, we propose a Mamba-based model to capture evolving graph dynamics by encoding long-range historical event streams and introducing a specialized module designed explicitly to encode edge-specific temporal patterns. This approach significantly differs from existing Mamba-based graph reasoning methods, enabling more effective temporal reasoning on CTDGs.

## 2.2 Preliminaries

**CTDG and Task Formulation.** We define CTDG and dynamic link prediction as follows.

**Definition 1** (Continuous-Time Dynamic Graph). *Let $\mathcal{N}$ and $\mathcal{T}$ denote a set of nodes and timestamps, respectively. A CTDG is a sequence of $|\mathcal{G}|$ chronological interactions $\mathcal{G} = \{(u_i, v_i, t_i)\}_{i=1}^{|\mathcal{G}|}$ with $0 \leq t_1 \leq t_2 \leq ... \leq t_{|\mathcal{G}|}$, where $u_i, v_i \in \mathcal{N}$ are the source and destination node of the $i$-th interaction happening at $t_i \in \mathcal{T}$, respectively. Each node $u \in \mathcal{N}$ can be equipped with a node feature $\mathbf{x}_u \in \mathbb{R}^{d_N}$, and each interaction $(u, v, t)$*

can be associated with a link (edge) feature $\mathbf{e}_{u,v}^t \in \mathbb{R}^{d_E}$. If $\mathcal{G}$ is not attributed, we set node and link features to zero vectors.

**Definition 2** (Dynamic Link Prediction). *Given a CTDG $\mathcal{G}$, a source node $u \in \mathcal{N}$, a destination node $v \in \mathcal{N}$, a timestamp $t \in \mathcal{T}$, and all the interactions before $t$, i.e., $\{(u_i, v_i, t_i)|t_i < t, (u_i, v_i, t_i) \in \mathcal{G}\}$, dynamic link prediction aims to predict whether the interaction $(u, v, t)$ exists.*

**S4 and Mamba SSM.** S4 and Mamba (Gu et al., 2022b; Gu & Dao, 2023) are inspired by a continuous system which can be described as $\mathbf{z}(\tau)' = \mathbf{A}\mathbf{z}(\tau) + \mathbf{B}q(\tau)$ and $r(\tau) = \mathbf{C}\mathbf{z}(\tau)$. $q(\tau) \in \mathbb{R}$ and $r(\tau) \in \mathbb{R}$ are the 1-dimensional input and output over time $\tau^2$, respectively. $\mathbf{A} \in \mathbb{R}^{d_1 \times d_1}, \mathbf{B} \in \mathbb{R}^{d_1 \times 1}, \mathbf{C} \in \mathbb{R}^{1 \times d_1}$ are three parameters deciding the system. Based on it, both S4 and Mamba include a time-scale parameter $\Delta \in \mathbb{R}$ and discretize all the parameters to adapt to a discretized system

$$\mathbf{z}_\tau = \bar{\mathbf{A}}\mathbf{z}_{\tau-1} + \bar{\mathbf{B}}p_\tau, \quad q_\tau = \mathbf{C}\mathbf{z}_\tau; \quad \bar{\mathbf{A}} = \exp(\Delta\mathbf{A}), \quad \bar{\mathbf{B}} = (\Delta\mathbf{A})^{-1}(\exp(\Delta\mathbf{A}) - \mathbf{I})\Delta\mathbf{B}. \tag{1}$$

Here, $\tau$ is also discretized to denote the position of a sequence element. Given Eq. 1 , sequence processing with S4 and Mamba can be written as computing an output sequence with convolution

$$\mathbf{q} = \mathbf{p} * \bar{\mathbf{K}}_{\text{SSM}}, \quad \text{where } \bar{\mathbf{K}}_{\text{SSM}} = [\mathbf{C}\bar{\mathbf{B}}, \mathbf{C}\bar{\mathbf{A}}\bar{\mathbf{B}}, ..., \mathbf{C}\bar{\mathbf{A}}^{|\mathbf{p}|-1}\bar{\mathbf{B}}] \in \mathbb{R}^{|\mathbf{p}|-1}. \tag{2}$$

$\mathbf{p} \in \mathbb{R}^{|\mathbf{p}|}$ and $\mathbf{q} \in \mathbb{R}^{|\mathbf{p}|}$ are input and output sequences, where $|\mathbf{p}|$ is the sequence length of $\mathbf{p}$. $*$ denotes the element-wise multiplication. When the dimension size of each element $p_\tau$ in $\mathbf{p}$ becomes higher (i.e., $p_\tau \in \mathbb{R}^{d_2}$ is a vector and $d_2 > 1$), both S4 and Mamba are in a Single-Input Single-Output (SISO) fashion, processing each input dimension in parallel with the same set of parameters. We follow Gu & Dao (2023) and denote the computation in Eq. 2 on the input sequences with vector elements as a function $\text{SSM}_{\bar{\mathbf{A}},\bar{\mathbf{B}},\mathbf{C}}(\cdot)^3$. Different from S4 which uses same parameters to process each element, Mamba changes its parameters into input-dependent by employing several trainable linear layers to map input into $\bar{\mathbf{B}}$, $\mathbf{C}$ and $\Delta$. The system is evolving as it processes different elements in the input sequence, making Mamba time-variant and suitable for modeling temporal sequences.

## 3 DyGMamba

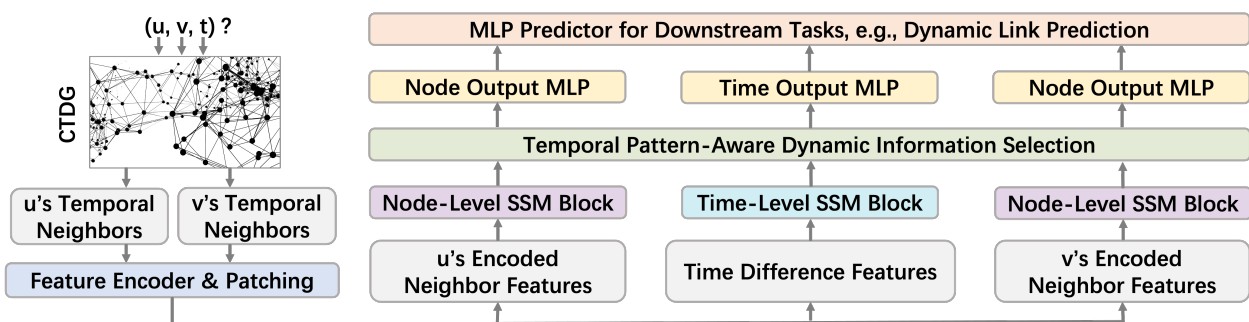

Figure 1: Model overview of DyGMamba.

Fig. 1 illustrates the overview of DyGMamba. Given a potential interaction $(u, v, t)$, CTDG models are asked to predict whether it exists or not. DyGMamba extracts the historical one-hop interactions of node $u$ and $v$ before timestamp $t$ from the CTDG $\mathcal{G}$ and gets two interaction sequences $\mathcal{S}_u^t = \{(u, u', t')|t' < t, (u, u', t') \in \mathcal{G}\} \cup \{(u', u, t')|t' < t, (u', u, t') \in \mathcal{G}\}$ and $\mathcal{S}_v^t = \{(v, v', t')|t' < t, (v, v', t') \in \mathcal{G}\} \cup \{(v', v, t')|t' < t, (v', v, t') \in \mathcal{G}\}$ containing $u$'s and $v$'s one-hop temporal neighbors $Nei_u^t = \{(u', t')|(u, u', t') \text{ or } (u', u, t') \in \mathcal{G}, t' < t\}$ and $Nei_v^t = \{(v', t')|(v, v', t') \text{ or } (v', v, t') \in \mathcal{G}, t' < t\}$ (link features are omitted for clarity). Then it encodes the neighbors in $Nei_u^t$ and $Nei_v^t$ to get two sequences of encoded neighbor representations for $u$ and $v$. To learn

---

[2] We use $\tau$ rather than $t$ to indicate time in a continuous system to distinguish from the time in CTDGs.

[3] Input and output of $\text{SSM}_{\bar{\mathbf{A}},\bar{\mathbf{B}},\mathbf{C}}(\cdot)$ are matrices where each row is a vector corresponding to an element. See App. H for more details of SISO and the function.

the edge-specific temporal pattern of $(u, v, t)$, we find the interactions between $u$ and $v$ before $t$, compute the time difference between each pair of neighboring interactions, and build a sequence of time differences $\mathcal{S}_{u,v}^t$. Finally, DyGMamba dynamically selects critical information by assigning different weights to different encoded neighbors based on the learned temporal pattern, and uses the selected information to achieve link prediction.

### 3.1 Learning One-Hop Temporal Neighbors

**Encode Neighbor Features.** Given one-hop temporal neighbors $Nei_u^t$ of the source node $u$, we sort them in the chronological order and append $(u, t)$ at the end to form a sequence of $Nei_u^t + 1$ temporal nodes. We take their node features from the dataset and stack them into a feature matrix $\tilde{\mathbf{X}}_u^t \in \mathbb{R}^{(|Nei_u^t|+1) \times d_N}$. Similarly, we build a link feature matrix $\tilde{\mathbf{E}}_u^t \in \mathbb{R}^{(|Nei_u^t|+1) \times d_E}$. To incorporate temporal information, we encode the time difference between $u$ and each one-hop temporal neighbor $(u', t')$ using the time encoding function introduced in TGAT (Xu et al., 2020): $\sqrt{1/d_T}[cos(\omega_1(t-t')+\phi_1), \ldots, cos(\omega_d(t-t')+\phi_{d_T})]$. $d_T$ is the dimension of time representation. $\omega_1 \ldots \omega_{d_T}$ and $\phi_1 \ldots \phi_{d_T}$ are trainable parameters. The time feature of $u$'s temporal neighbors are denoted as $\tilde{\mathbf{T}}_u^t \in \mathbb{R}^{(|Nei_u^t|+1) \times d_T}$. We follow the same way to get $\tilde{\mathbf{X}}_v^t \in \mathbb{R}^{(|Nei_v^t|+1) \times d_N}$, $\tilde{\mathbf{E}}_v^t \in \mathbb{R}^{(|Nei_v^t|+1) \times d_E}$ and $\tilde{\mathbf{T}}_v^t \in \mathbb{R}^{(|Nei_v^t|+1) \times d_T}$ for $v$'s temporal neighbors. Following Tian et al. (2024), we also consider the historical node interaction frequencies in the interaction sequences $\mathcal{S}_u^t$ and $\mathcal{S}_v^t$ of source $u$ and destination $v$. For example, assume the interacted nodes of $u$ and $v$ (arranged in chronological order) are $\{a, v, a\}$ and $\{b, b, u, a\}$, the appearing frequencies of $a$, $b$ in $u/v$'s historical interactions are $2/1$, $0/2$, respectively. And the frequency of the interaction involving $u$ and $v$ is 1. Thus, the node interaction frequency features of $u$ and $v$ are written as $\tilde{F}_u^t = [[2,1],[1,1],[2,1],[0,1]]^\top$ and $\tilde{F}_v^t = [[0,2],[0,2],[1,1],[2,1],[0,1]]^\top$, respectively. Note that the last elements ($[0,1]$ and $[0,1]$) in $\tilde{F}_u^t$ and $\tilde{F}_v^t$ correspond to the appended $(u, t)$ and $(v, t)$ not existing in the observed histories. We initialize them with $[0, \textit{number of historical interactions between } u, v]$. An encoding multilayer perceptron (MLP) $f(\cdot) : \mathbb{R} \to \mathbb{R}^{d_F}$ is employed to encode these features into representations: $\tilde{\mathbf{F}}_u^t = f(\tilde{F}_u^t[:, 0]) + f(\tilde{F}_u^t[:, 1]) \in \mathbb{R}^{(|Nei_u^t|+1) \times d_F}$, $\tilde{\mathbf{F}}_v^t = f(\tilde{F}_v^t[:, 0]) + f(\tilde{F}_v^t[:, 1]) \in \mathbb{R}^{(|Nei_v^t|+1) \times d_F}$.

**Patching Neighbors.** We employ the patching technique proposed by (Yu et al., 2023) to save computational resources when dealing with a large number of temporal neighbors. We treat $p$ temporally adjacent neighbors as a patch and flatten their features. For example, with patching, $\tilde{\mathbf{X}}_u^t \in \mathbb{R}^{(|Nei_u^t|+1) \times d_N}$ results in a new patched feature matrix $\mathbf{X}_u^t \in \mathbb{R}^{\lceil(|Nei_u^t|+1)/p\rceil \times (p \cdot d_N)}$ (we pad $\tilde{\mathbf{X}}_u^t$ with zero-valued features when $|Nei_u^t| + 1$ cannot be divided by $p$). Similarly, we get $\mathbf{E}_\theta^t \in \mathbb{R}^{\lceil(|Nei_\theta^t|+1)/p\rceil \times (p \cdot d_E)}$, $\mathbf{T}_\theta^t \in \mathbb{R}^{\lceil(|Nei_\theta^t|+1)/p\rceil \times (p \cdot d_T)}$ and $\mathbf{F}_\theta^t \in \mathbb{R}^{\lceil(|Nei_\theta^t|+1)/p\rceil \times (p \cdot d_F)}$ ($\theta$ is either $u$ or $v$). Each row of a feature matrix corresponds to an element of the input sequence sent into an SSM later. Recall that SSMs process sequences in a recurrent way. Patching decreases the length of the sequence by roughly $p$ times, making great contribution in saving computational resources.

**Node-Level SSM Block.** We first map the padded features of $u$'s and $v$'s one-hop temporal neighbors to the same dimension $d$, i.e., $\mathbf{X}_\theta^t := f_N(\mathbf{X}_\theta^t)$, $\mathbf{E}_\theta^t := f_E(\mathbf{E}_\theta^t)$, $\mathbf{T}_\theta^t := f_T(\mathbf{T}_\theta^t)$, $\mathbf{F}_\theta^t := f_F(\mathbf{F}_\theta^t)$. $f_N(\cdot) : \mathbb{R}^{p \cdot d_N} \to \mathbb{R}^d$, $f_E(\cdot) : \mathbb{R}^{p \cdot d_E} \to \mathbb{R}^d$, $f_T(\cdot) : \mathbb{R}^{p \cdot d_T} \to \mathbb{R}^d$, $f_F(\cdot) : \mathbb{R}^{p \cdot d_F} \to \mathbb{R}^d$ are four MLPs for different types of neighbor features. We take the concatenation of them as the encoded representations of the temporal neighbors, i.e., $\mathbf{H}_\theta^t = \mathbf{X}_\theta^t \| \mathbf{E}_\theta^t \| \mathbf{T}_\theta^t \| \mathbf{F}_\theta^t \in \mathbb{R}^{\lceil(|Nei_\theta^t|+1)/p\rceil \times 4d}$. We input $\mathbf{H}_u^t$ and $\mathbf{H}_v^t$ separately into a node-level SSM block to learn the temporal dependencies of temporal neighbors. The node-level SSM block consists of $l_N$ layers, where each layer is defined as follows (Eq. 3-4). First, we input $\mathbf{H}_\theta^t$ into a Mamba SSM

$$\mathbf{B}_1 = \mathbf{H}_\theta^t \mathbf{W}_{\mathbf{B}_1} \in \mathbb{R}^{\lceil(|Nei_\theta^t|+1)/p\rceil \times d_{\text{SSM}}}, \quad \mathbf{C}_1 = \mathbf{H}_\theta^t \mathbf{W}_{\mathbf{C}_1} \in \mathbb{R}^{\lceil(|Nei_\theta^t|+1)/p\rceil \times d_{\text{SSM}}}; \tag{3a}$$

$$\Delta_1 = \text{Softplus}(\text{Broadcast}_{4d}(\mathbf{H}_\theta^t \mathbf{W}_{\Delta_1}) + \text{Param}_{\Delta_1}) \in \mathbb{R}^{\lceil(|Nei_\theta^t|+1)/p\rceil \times 4d}; \tag{3b}$$

$$\bar{\mathbf{A}}_1 = \exp(\Delta_1 \mathbf{A}_1), \quad \bar{\mathbf{B}}_1 = (\Delta_1 \mathbf{A}_1)^{-1}(\exp(\Delta_1 \mathbf{A}_1) - \mathbf{I})\Delta_1 \mathbf{B}_1; \tag{3c}$$

$$\mathbf{H}_\theta^t := \mathbf{H}_\theta^t + \text{SSM}_{\bar{\mathbf{A}}_1, \bar{\mathbf{B}}_1, \mathbf{C}_1}(\mathbf{H}_\theta^t). \tag{3d}$$

$\mathbf{W}_{\mathbf{B}_1}, \mathbf{W}_{\mathbf{C}_1} \in \mathbb{R}^{4d \times d_{\text{SSM}}}$ and $\mathbf{W}_{\Delta_1} \in \mathbb{R}^{4d \times 1}$. $\bar{\mathbf{A}}_1, \bar{\mathbf{B}}_1 \in \mathbb{R}^{\lceil(|Nei_\theta^t|+1)/p\rceil \times 4d \times d_{\text{SSM}}}$ are discretized parameters. $\text{Param}_{\Delta_1} \in \mathbb{R}^{\lceil(|Nei_\theta^t|+1)/p\rceil \times 4d}$ is a parameter defined by Gu & Dao (2023). $\text{Broadcast}_{4d}(\cdot)$ is a function that

copies its vector input for $4d$ times to form a matrix with $4d$ identical columns (following the definition in Gu & Dao (2023) ). $\mathbf{I}$ is an identity matrix. Then we use an MLP $f_{\text{node}}(\cdot) : \mathbb{R}^{4d} \to \mathbb{R}^{4d}$ on SSM's output

$$\mathbf{H}_\theta^t := \mathbf{H}_\theta^t + f_{\text{node}}\left(\text{LayerNorm}(\mathbf{H}_\theta^t)\right). \tag{4}$$

After $l_N$ layers, we have $\mathbf{H}_u^t$ and $\mathbf{H}_v^t$ that contain the encoded information of all one-hop temporal neighbors for the entities $u$ and $v$ as well as the information of themselves. Since we sort temporal neighbors chronologically, our node-level SSM block can directly learn the temporal dynamics for graph forecasting.

## 3.2  Learning from Edge-Specific Temporal Patterns

**Time-Level SSM Block.**  To capture edge-specific temporal patterns, we use another time-level SSM block consisting of $l_T$ layers. We first find out $k$ temporally nearest historical interactions between $u$ and $v$ before $t$ and sort them in the chronological order, i.e., $\{(u,v,t_0),...,(u,v,t_{k-1})|t_0 < ... < t_{k-1} < t\}$. Then we construct a timestamp sequence $\{t_0, t_1, ..., t_{k-1}, t\}$ based on these interactions and the prediction timestamp $t$. We compute the time difference between each neighboring pair of them and further get a time difference sequence $\{t_1 - t_0, t_2 - t_1, ..., t - t_{k-1}\}$, representing the change of time interval between two identical interactions. Each element in this sequence is input into the time encoding function stated above to get a edge-specific (specific to the edge $(u,v,t)$) time feature. The features are stacked into a feature matrix $\mathbf{H}_{u,v}^t \in \mathbb{R}^{k \times d_T}$ and mapped by an MLP $f_{\text{map1}}(\cdot) : \mathbb{R}^{d_T} \to \mathbb{R}^{\gamma d}$ ($\gamma \in [0,1]$ is a hyperparameter), i.e., $\mathbf{H}_{u,v}^t := f_{\text{map1}}(\mathbf{H}_{u,v}^t)$. A time-level SSM layer takes $\mathbf{H}_{u,v}^t$ as input and computes

$$\mathbf{B}_2 = \mathbf{H}_{u,v}^t \mathbf{W}_{\mathbf{B}_2} \in \mathbb{R}^{k \times d_{\text{SSM}}}, \quad \mathbf{C}_2 = \mathbf{H}_{u,v}^t \mathbf{W}_{\mathbf{C}_2} \in \mathbb{R}^{k \times d_{\text{SSM}}}; \tag{5a}$$

$$\Delta_2 = \text{Softplus}(\text{Broadcast}_{\gamma d}(\mathbf{H}_{u,v}^t \mathbf{W}_{\Delta_2}) + \text{Param}_{\Delta_2}) \in \mathbb{R}^{k \times \gamma d}; \tag{5b}$$

$$\bar{\mathbf{A}}_2 = \exp(\Delta_2 \mathbf{A}_2), \quad \bar{\mathbf{B}}_2 = (\Delta_2 \mathbf{A}_2)^{-1}(\exp(\Delta_2 \mathbf{A}_2) - \mathbf{I})\Delta_2 \mathbf{B}_2; \tag{5c}$$

$$\mathbf{H}_{u,v}^t := \mathbf{H}_{u,v}^t + \text{SSM}_{\bar{\mathbf{A}}_2, \bar{\mathbf{B}}_2, \mathbf{C}_2}(\mathbf{H}_{u,v}^t). \tag{5d}$$

$\mathbf{W}_{\mathbf{B}_2}, \mathbf{W}_{\mathbf{C}_2} \in \mathbb{R}^{\gamma d \times d_{\text{SSM}}}$ and $\mathbf{W}_{\Delta_2} \in \mathbb{R}^{\gamma d \times 1}$. $\bar{\mathbf{A}}_2, \bar{\mathbf{B}}_2 \in \mathbb{R}^{k \times \gamma d \times d_{\text{SSM}}}$ are discretized parameters. $\text{Param}_{\Delta_2} \in \mathbb{R}^{k \times \gamma d}$ is a parameter defined as same as $\text{Param}_{\Delta_1}$. In practice, we set $k$ to a number much smaller than $|Nei_\theta^t|$, e.g., 10. This ensures that time-level SSM will not incur huge computational burden and the model focuses more on the recent histories. Note that we cannot always find $k$ recent historical interactions between each pair of nodes, leading to varying lengths of time difference sequences for different $(u,v,t)$ in a batch of data. To enable batch processing, we set the time difference without a found historical interaction to a very large number $10^{10}$. For example, if $k = 2$, and for $(u,v,t)$ we can only find $(u,v,t_0)$. The time difference sequence will be $\{10^{10}, t - t_0\}$. $10^{10}$ is much larger than $t - t_0$, indicating that $u$ and $v$ have not had an interaction for an extremely long time, same as existing no historical interaction. We further explain why we use SSM to learn temporal patterns in App. I.

**Dynamic Information Selection with Temporal Patterns.**  After the time-level SSM block, we compute a compressed representation to represent the edge-specific temporal pattern by averaging over $k$ encoded time intervals: $\mathbf{h}_{u,v}^t = \text{MeanPooling}(\mathbf{H}_{u,v}^t)$. As a result, we have $\mathbf{h}_{u,v}^t \in \mathbb{R}^{\gamma d}$ to represent the temporal pattern specific to the edge $(u,v,t)$. To leverage learned temporal pattern, we use it to dynamically select the information from the encoded temporal neighbors $\mathbf{H}_\theta^t$

$$\hat{\mathbf{h}}_{u,v}^t = f_{\text{map2}}(\mathbf{h}_{u,v}^t) \in \mathbb{R}^{4d}; \tag{6a}$$

$$\hat{\mathbf{h}}_\theta^t = \mathbf{w}_{\text{agg}}^\top \mathbf{H}_\theta^t \in \mathbb{R}^{4d}, \text{ where } \mathbf{w}_{\text{agg}} = f_{\text{map3}}(\mathbf{H}_\theta^t) \in \mathbb{R}^{\lceil(|Nei_\theta^t|+1)/p\rceil}; \tag{6b}$$

$$\alpha_u = f'(\hat{\mathbf{h}}_v^t) * \hat{\mathbf{h}}_{u,v}^t \in \mathbb{R}^{4d}, \ \alpha_v = f'(\hat{\mathbf{h}}_u^t) * \hat{\mathbf{h}}_{u,v}^t \in \mathbb{R}^{4d}; \tag{6c}$$

$$\mathbf{h}_\theta^t = \beta_\theta^\top \mathbf{H}_\theta^t, \text{ where } \beta_\theta = \text{Softmax}(\mathbf{H}_\theta^t \alpha_\theta) \in \mathbb{R}^{\lceil(|Nei_\theta^t|+1)/p\rceil}. \tag{6d}$$

$f_{\text{map2}}(\cdot) : \mathbb{R}^{\gamma d} \to \mathbb{R}^{4d}$ and $f_{\text{map3}}(\cdot) : \mathbb{R}^{4d} \to \mathbb{R}^1$ are two mapping MLPs. $f'(\cdot) : \mathbb{R}^{4d} \to \mathbb{R}^{4d}$ is another MLP introducing training parameters. Note that $\alpha_u/\alpha_v$ is computed by considering both the edge-specific temporal pattern and the opposite node $v/u$. In the node-level SSM block, we separately model the one-hop

temporal neighbors of each node $\theta$, making it hard to connect $u$ and $v$. Computing $\alpha_\theta$ as Eq. 6c helps to strengthen the connection between both nodes and meanwhile incorporates the learned temporal pattern. $\beta_\theta$ is derived by transforming the queried results based on $\alpha_\theta$ into weights. It is then used to compute a weighted-sum of all temporal neighbors for representing $\theta$ at $t$, i.e., $\mathbf{h}_\theta^t$. The neighbors assigned with greater weights from $\beta_\theta$ are selected as more critical and will contribute more to $\mathbf{h}_\theta^t$. Finally, we output the representations of $u$, $v$ and the edge-specific temporal pattern by employing two output MLPs $f_{\text{out1}}(\cdot) : \mathbb{R}^{4d} \to \mathbb{R}^{d_N}$ and $f_{\text{out2}}(\cdot) : \mathbb{R}^{\gamma d} \to \mathbb{R}^{d_N}$, i.e., $\mathbf{h}_\theta^t := f_{\text{out1}}(\mathbf{h}_\theta^t) \in \mathbb{R}^{d_N}$, $\mathbf{h}_{u,v}^t := f_{\text{out2}}(\mathbf{h}_{u,v}^t) \in \mathbb{R}^{d_N}$.

### 3.3 Leveraging Learned Representations for Link Prediction

We leverage $\mathbf{h}_\theta^t$ and $\mathbf{h}_{u,v}^t$ for dynamic link prediction. We employ a prediction MLP, i.e., $f_{\text{LP}}(\cdot) : \mathbb{R}^{3d_N} \to \mathbb{R}$, as the predictor. The probability of existing a link $(u, v, t)$ is computed as $y'(u, v, t) = \text{Sigmoid}(f_{\text{LP}}(\mathbf{h}_u^t \| \mathbf{h}_v^t \| \mathbf{h}_{u,v}^t))$. For model parameter learning, we use the following loss function

$$\mathcal{L} = -\frac{1}{2M} \sum_{2M} \left( y(u, v, t) \log(y'(u, v, t)) + (1 - y(u, v, t)) \log(1 - y'(u, v, t)) \right). \tag{7}$$

$y(u, v, t)$ is the ground truth label denoting the existence of $(u, v, t)$ (1/0 means existing/non-existing). $M$ is the total number of edges existing in the training data (positive edges). We follow previous work (Yu et al., 2023) and randomly sample one negative edge for each positive edge during training. Therefore, in total we have $2M$ edges considered in our loss $\mathcal{L}$.

## 4 Experiments

In Sec. 4.2.1, we validate DyGMamba's ability in CTDG representation learning by comparing it with baseline methods on dynamic link prediction[4]. We show the effectiveness of model components by conducting ablation studies (Sec. 4.2.2) and analysis on synthetic datasets (Sec. 4.2.3). In Sec. 4.3.1, we show DyGMamba's efficiency against various baselines. We also show that it achieves much stronger scalability in modeling long-term temporal information compared with the current state-of-the-art DyGFormer (Sec. 4.3.2 and Sec. 4.3.3).

### 4.1 Experimental Setting

**CTDG Datasets and Baselines.** We consider seven real-world CTDG datasets collected by (Poursafaei et al., 2022), i.e., LastFM, Enron, MOOC, Reddit, Wikipedia, UCI and Social Evo.. Dataset statistics are presented in App. A.1. Among them, we take LastFM, Enron and MOOC as long-range temporal dependent datasets because according to Yu et al. (2023), much longer histories are needed for optimal representation learning on them. We compare DyGMamba with ten recent CTDG baseline models, i.e., JODIE (Kumar et al., 2019), DyRep (Trivedi et al., 2019), TGAT (Xu et al., 2020), TGN (Rossi et al., 2020), CAWN (Wang et al., 2021c), EdgeBank (Poursafaei et al., 2022), TCL (Wang et al., 2021a), GraphMixer (Cong et al., 2023), DyGFormer (Yu et al., 2023) and CTAN (Gravina et al., 2024). Among them, only DyGFormer and CTAN are designed for long-range temporal information propagation. Detailed descriptions of baseline methods are presented in App. B. We also implemented FreeDyG (Tian et al., 2024) by using its official code repository, however, on LastFM, we find that FreeDyG's loss cannot converge and the reported results are not reproducible. So we do not report its performance in our paper.

**Implementation Details and Evaluation Settings.** We use the implementations and the best hyper-parameters provided by Yu et al. (2023) for all baseline models except CTAN. For CTAN, we use its official implementation, fixing the number of layers to 5. All models are trained with a batch size of 200 for fair efficiency analysis. For DyGMamba, we report the number of sampled one-hop temporal neighbors $\rho$ and the patch size $p$ here. On Wikipedia, Social Evo., and UCI, $\rho$ & $p$ = 32 & 1. On Reddit, $\rho$ & $p$ = 64 &

---

[4]To supplement, we also validate on the dynamic node classification task. Since current mainstream datasets of this task requires no long-term temporal reasoning, we put the discussion in App. F. Additionally, we also benchmark DyGMamba on DTDGs in App. K. This serves as supplementary experiment and does not directly connect to our main focus.

2. On MOOC, $\rho$ & $p$ = 128 & 4. On Enron, $\rho$ & $p$ = 256 & 8. On LastFM, $\rho$ & $p$ = 512 & 16. Note that to fairly compare DyGMamba's efficiency with DyGFormer, we keep the sequence length $\rho/p$ input into the SSM as same as the length input into Transformer in Yu et al. (2023), i.e., $\rho/p = 32$. All experiments are implemented with PyTorch (Paszke et al., 2019) on a server equipped with an AMD EPYC 7513 32-Core Processor and a single NVIDIA A40 with 45GB memory. We run each experiment for five times with five random seeds and report the mean results together with error bars. Further implementation details including complete hyperparameter configurations are presented in App. C. We employ two evaluation settings following previous works: the transductive and inductive settings. As suggested in (Poursafaei et al., 2022), we do link prediction evaluation using three negative sampling strategies (NSSs): random, historical and inductive. Historical NSS is only considered under the transductive setting. See App. D for detailed explanations. We employ two metrics, i.e., average precision (AP) and area under the receiver operating characteristic curve (AUC-ROC)

Table 1: AP of transductive dynamic link prediction. The best and the second best results are marked as **bold** and underlined, respectively. CTAN cannot be trained before 120 hours timeout on Social Evo. so is ranked bottom on this dataset.

| NSS | Datasets | JODIE | DyRep | TGAT | TGN | CAWN | EdgeBank | TCL | GraphMixer | DyGFormer | CTAN | DyGMamba |
|---|---|---|---|---|---|---|---|---|---|---|---|---|
| Random | LastFM | 70.95 ± 2.94 | 71.85 ± 2.44 | 73.30 ± 0.18 | 75.31 ± 5.62 | 86.60 ± 0.11 | 79.29 ± 0.19 | 76.62 ± 1.83 | 75.56 ± 0.19 | 92.95 ± 0.14 | 86.44 ± 0.80 | **93.35 ± 0.20** |
| | Enron | 84.85 ± 3.13 | 79.80 ± 2.28 | 70.76 ± 1.05 | 86.98 ± 1.05 | 89.50 ± 0.10 | 83.53 ± 0.00 | 85.41 ± 0.71 | 82.13 ± 0.30 | 92.42 ± 0.11 | 92.52 ± 1.20 | **92.65 ± 0.12** |
| | MOOC | 81.04 ± 0.83 | 81.50 ± 0.77 | 85.71 ± 0.20 | 89.15 ± 1.69 | 80.30 ± 0.43 | 57.97 ± 0.00 | 83.89 ± 0.86 | 82.80 ± 0.15 | 87.66 ± 0.48 | 84.71 ± 2.85 | **89.21 ± 0.08** |
| | Reddit | 98.31 ± 0.06 | 98.18 ± 0.03 | 98.57 ± 0.01 | 98.65 ± 0.04 | 99.11 ± 0.01 | 94.86 ± 0.00 | 97.78 ± 0.02 | 97.31 ± 0.01 | 99.22 ± 0.01 | 97.21 ± 0.84 | **99.32 ± 0.01** |
| | Wikipedia | 96.51 ± 0.22 | 94.88 ± 0.29 | 96.88 ± 0.06 | 98.45 ± 0.10 | 98.77 ± 0.01 | 90.37 ± 0.00 | 97.75 ± 0.04 | 97.22 ± 0.02 | 99.03 ± 0.03 | 96.61 ± 0.79 | **99.15 ± 0.02** |
| | UCI | 89.28 ± 1.02 | 66.11 ± 2.75 | 79.40 ± 0.61 | 92.33 ± 0.64 | 95.13 ± 0.23 | 76.20 ± 0.00 | 86.63 ± 1.30 | 93.15 ± 0.41 | 95.74 ± 0.17 | 76.64 ± 4.11 | **95.91 ± 0.15** |
| | Social Evo. | 89.88 ± 0.40 | 88.39 ± 0.69 | 93.33 ± 0.06 | 93.45 ± 0.29 | 84.90 ± 0.11 | 74.95 ± 0.00 | 93.82 ± 0.19 | 93.36 ± 0.06 | 94.63 ± 0.07 | Timeout | **94.77 ± 0.01** |
| | **Avg. Rank** | 8.29 | 9.29 | 7.00 | 4.29 | 6.00 | 9.43 | 5.57 | 6.43 | 2.43 | 6.29 | **1.00** |
| Historical | LastFM | 74.38 ± 6.27 | 71.85 ± 2.91 | 71.60 ± 0.36 | 75.03 ± 6.90 | 69.93 ± 0.33 | 73.03 ± 0.00 | 71.02 ± 2.07 | 72.28 ± 0.37 | 81.51 ± 0.14 | 82.29 ± 0.94 | **83.02 ± 0.16** |
| | Enron | 69.13 ± 1.66 | 72.58 ± 1.83 | 64.24 ± 1.24 | 74.31 ± 0.99 | 65.40 ± 0.36 | 76.53 ± 0.00 | 72.39 ± 0.61 | 77.35 ± 1.22 | 76.93 ± 0.76 | 77.24 ± 1.53 | **77.77 ± 1.32** |
| | MOOC | 78.62 ± 2.43 | 75.14 ± 2.86 | 82.83 ± 0.71 | 85.46 ± 2.32 | 74.46 ± 0.53 | 60.71 ± 0.00 | 78.51 ± 1.24 | 77.09 ± 0.83 | 85.65 ± 0.89 | 67.73 ± 2.08 | **85.89 ± 0.94** |
| | Reddit | 79.96 ± 0.30 | 79.40 ± 0.30 | 79.78 ± 0.25 | 81.05 ± 0.32 | 80.96 ± 0.28 | 73.59 ± 0.00 | 77.38 ± 0.20 | 78.39 ± 0.40 | 81.63 ± 1.08 | **89.77 ± 2.28** | 81.80 ± 1.52 |
| | Wikipedia | 81.16 ± 0.73 | 79.46 ± 0.95 | 87.31 ± 0.36 | 87.31 ± 0.25 | 66.77 ± 6.62 | 73.35 ± 0.00 | 86.12 ± 1.69 | 90.74 ± 0.06 | 70.13 ± 11.02 | **95.91 ± 0.10** | 81.77 ± 1.09 |
| | UCI | 74.77 ± 5.35 | 55.89 ± 2.83 | 66.78 ± 0.77 | 81.32 ± 1.26 | 64.69 ± 1.78 | 65.50 ± 0.00 | 74.62 ± 2.70 | **83.88 ± 1.06** | 80.44 ± 1.16 | 76.62 ± 0.33 | 81.03 ± 1.09 |
| | Social Evo. | 91.26 ± 2.47 | 92.86 ± 0.90 | 95.31 ± 0.30 | 93.84 ± 1.68 | 85.65 ± 0.11 | 80.57 ± 0.00 | 95.93 ± 0.63 | 95.30 ± 0.34 | 97.05 ± 0.16 | Timeout | **97.35 ± 0.52** |
| | **Avg. Rank** | 6.57 | 8.14 | 6.57 | 4.14 | 9.29 | 8.71 | 7.00 | 4.71 | 4.00 | 4.71 | **2.14** |
| Inductive | LastFM | 62.63 ± 6.89 | 62.49 ± 3.04 | 71.16 ± 0.33 | 65.09 ± 7.05 | 67.38 ± 0.57 | 75.49 ± 0.00 | 62.76 ± 0.81 | 67.87 ± 0.37 | 72.60 ± 0.06 | **80.06 ± 0.85** | 73.63 ± 0.54 |
| | Enron | 69.51 ± 1.06 | 66.78 ± 2.21 | 63.16 ± 0.59 | 73.27 ± 0.58 | 75.08 ± 0.81 | 73.89 ± 0.00 | 70.98 ± 0.96 | 74.12 ± 0.65 | 78.22 ± 0.80 | 72.02 ± 2.64 | **80.86 ± 1.24** |
| | MOOC | 66.56 ± 1.49 | 61.48 ± 0.96 | 76.96 ± 0.89 | 77.59 ± 1.83 | 73.55 ± 0.36 | 49.43 ± 0.00 | 76.35 ± 1.41 | 74.24 ± 0.75 | 80.99 ± 0.88 | 64.93 ± 3.31 | **81.11 ± 0.63** |
| | Reddit | 86.93 ± 0.21 | 86.06 ± 0.36 | 89.93 ± 0.10 | 88.12 ± 0.13 | **91.89 ± 0.18** | 85.48 ± 0.00 | 86.97 ± 0.26 | 85.37 ± 0.26 | 91.06 ± 0.60 | 90.99 ± 2.19 | 91.15 ± 0.54 |
| | Wikipedia | 74.78 ± 0.56 | 70.55 ± 1.22 | 86.77 ± 0.29 | 85.80 ± 0.15 | 69.27 ± 7.07 | 80.63 ± 0.00 | 72.54 ± 4.69 | 88.54 ± 0.20 | 62.00 ± 14.00 | **94.15 ± 0.08** | 79.86 ± 2.18 |
| | UCI | 66.02 ± 1.28 | 54.64 ± 2.52 | 67.63 ± 0.51 | 70.34 ± 0.72 | 64.08 ± 1.06 | 57.43 ± 0.00 | 73.49 ± 2.21 | **79.57 ± 0.61** | 70.51 ± 1.83 | 66.25 ± 0.51 | 71.95 ± 2.51 |
| | Social Evo. | 91.08 ± 3.29 | 92.84 ± 0.98 | 95.20 ± 0.30 | 94.58 ± 1.52 | 88.50 ± 0.13 | 83.69 ± 0.00 | 96.14 ± 0.63 | 95.11 ± 0.32 | 97.62 ± 0.12 | Timeout | **97.68 ± 0.42** |
| | **Avg. Rank** | 8.29 | 9.57 | 5.43 | 5.43 | 6.57 | 7.57 | 6.00 | 5.00 | 4.00 | 5.71 | **2.43** |

Table 2: AP of inductive dynamic link prediction. EdgeBank cannot do inductive link prediction so is not reported.

| NSS | Datasets | JODIE | DyRep | TGAT | TGN | CAWN | TCL | GraphMixer | DyGFormer | CTAN | DyGMamba |
|---|---|---|---|---|---|---|---|---|---|---|---|
| Random | LastFM | 83.13 ± 1.19 | 83.47 ± 1.06 | 78.40 ± 0.30 | 81.18 ± 3.27 | 89.33 ± 0.06 | 81.38 ± 1.53 | 82.07 ± 0.31 | 94.17 ± 0.10 | 60.40 ± 3.01 | **94.42 ± 0.21** |
| | Enron | 78.97 ± 1.59 | 73.97 ± 3.00 | 66.67 ± 1.07 | 78.76 ± 1.69 | 86.30 ± 0.56 | 82.61 ± 0.61 | 75.55 ± 0.81 | 89.62 ± 0.27 | 74.61 ± 1.64 | **89.67 ± 0.27** |
| | MOOC | 80.57 ± 0.52 | 80.50 ± 0.68 | 85.28 ± 0.30 | 88.01 ± 1.48 | 81.32 ± 0.42 | 82.28 ± 0.99 | 81.38 ± 0.17 | 87.05 ± 0.51 | 64.99 ± 2.24 | **88.64 ± 0.08** |
| | Reddit | 96.43 ± 0.16 | 95.89 ± 0.26 | 97.13 ± 0.04 | 97.41 ± 0.12 | 98.62 ± 0.01 | 95.01 ± 0.10 | 95.24 ± 0.08 | 98.83 ± 0.02 | 80.07 ± 2.53 | **98.97 ± 0.01** |
| | Wikipedia | 94.91 ± 0.32 | 92.21 ± 0.29 | 96.26 ± 0.12 | 97.81 ± 0.18 | 98.27 ± 0.02 | 97.48 ± 0.06 | 96.61 ± 0.04 | 98.58 ± 0.01 | 93.58 ± 0.65 | **98.77 ± 0.03** |
| | UCI | 79.73 ± 1.48 | 58.39 ± 2.38 | 79.10 ± 0.49 | 87.81 ± 1.32 | 92.61 ± 0.35 | 84.19 ± 1.37 | 91.17 ± 0.29 | 94.45 ± 0.13 | 49.78 ± 5.02 | **94.76 ± 0.19** |
| | Social Evo. | 91.72 ± 0.66 | 89.10 ± 1.90 | 91.47 ± 0.10 | 90.74 ± 1.40 | 79.83 ± 0.14 | 92.51 ± 0.11 | 91.89 ± 0.05 | 93.05 ± 0.10 | Timeout | **93.13 ± 0.05** |
| | **Avg. Rank** | 6.29 | 8.00 | 7.00 | 5.14 | 4.43 | 5.57 | 5.86 | 2.14 | 9.57 | 1.00 |
| Inductive | LastFM | 71.37 ± 3.45 | 69.75 ± 2.73 | 76.26 ± 0.34 | 68.47 ± 6.07 | 71.28 ± 0.43 | 68.79 ± 0.93 | 76.27 ± 0.37 | 75.07 ± 1.45 | 55.60 ± 3.91 | **76.76 ± 0.43** |
| | Enron | 66.99 ± 1.15 | 62.64 ± 2.33 | 59.95 ± 1.00 | 64.51 ± 1.66 | 60.61 ± 0.63 | 68.93 ± 1.34 | **71.71 ± 1.33** | 67.21 ± 0.72 | 68.66 ± 2.31 | 68.77 ± 0.60 |
| | MOOC | 64.67 ± 1.18 | 62.05 ± 2.11 | 77.43 ± 0.81 | 76.81 ± 2.83 | 74.36 ± 0.78 | 75.95 ± 1.46 | 73.87 ± 0.99 | 80.66 ± 0.94 | 57.49 ± 1.34 | **80.75 ± 1.00** |
| | Reddit | 62.54 ± 0.52 | 61.07 ± 0.86 | 63.96 ± 0.25 | 65.27 ± 0.57 | 64.10 ± 0.22 | 61.45 ± 0.25 | 64.82 ± 0.30 | 65.03 ± 1.20 | **78.35 ± 5.03** | 65.30 ± 1.05 |
| | Wikipedia | 68.22 ± 0.36 | 61.07 ± 0.82 | 84.19 ± 0.96 | 81.96 ± 0.62 | 62.34 ± 6.79 | 71.46 ± 4.95 | 87.47 ± 0.25 | 57.90 ± 11.05 | **92.61 ± 0.90** | 71.14 ± 2.44 |
| | UCI | 63.57 ± 2.15 | 52.63 ± 1.87 | 69.77 ± 0.43 | 69.94 ± 0.50 | 63.44 ± 1.52 | 74.39 ± 1.81 | **81.40 ± 0.52** | 70.25 ± 2.02 | 52.31 ± 2.67 | 72.17 ± 2.20 |
| | Social Evo. | 89.06 ± 1.23 | 87.30 ± 1.55 | 94.24 ± 0.36 | 90.67 ± 2.41 | 80.30 ± 0.21 | 95.94 ± 0.37 | 94.56 ± 0.24 | 96.73 ± 0.11 | Timeout | **96.83 ± 0.56** |
| | **Avg. Rank** | 6.86 | 8.57 | 5.29 | 5.43 | 7.43 | 4.86 | 3.14 | 4.43 | 6.57 | **2.43** |

## 4.2 Performance Analysis

### 4.2.1 Comparative Study on Benchmark Datasets

We report the AP of transductive and inductive link prediction in Table 1 and 2 (AUC-ROC reported in Table 12 and 13 in App. E). We find that: (1) DyGMamba constantly ranks top 1 under the random NSS,

showing a superior performance; (2) Under the historical and inductive NSS, DyGMamba can achieve the best average rank compared with all baselines. More importantly, it shows more superiority on the datasets where encoding longer-term temporal dependencies is necessary, e.g., on LastFM, Enron and MOOC. (3) Among the models that can do long range propagation of information over time (i.e., DyGFormer, CTAN and DyGMamba), DyGMamba achieves the best average rank under any NSS setting in both transductive and inductive link prediction. On the long-range temporal dependent datasets, DyGMamba outperforms DyGFormer and CTAN in most cases; (4) CTAN achieves much better results in transductive than in inductive link prediction. This is because CTAN requires multi-hop temporal neighbors to learn node representations, which is difficult for unseen nodes. By contrast, DyGMamba and DyGFormer require only one-hop temporal neighbors, thus performing much better in inductive link prediction.

Table 3: Ablation studies under transductive setting. R/H/I means random/historical/inductive NSS. Metric is AP.

| Datasets | LastFM | | | Enron | | | MOOC | | | Reddit | | | Wikipedia | | | UCI | | | Social Evo. | | |
|---|---|---|---|---|---|---|---|---|---|---|---|---|---|---|---|---|---|---|---|---|---|
| Models | R | H | I | R | H | I | R | H | I | R | H | I | R | H | I | R | H | I | R | H | I |
| Variant A | 93.14 | 80.30 | 71.29 | 91.35 | 70.07 | 75.44 | 87.78 | 83.25 | 77.04 | 99.19 | 81.60 | 90.70 | 98.99 | 80.99 | 79.26 | 94.88 | 79.37 | 70.43 | 94.59 | 96.97 | 97.42 |
| Variant B | 93.07 | 82.53 | 72.97 | 92.46 | 76.88 | 78.87 | 86.95 | 83.78 | 75.81 | 97.97 | 73.47 | 84.16 | 94.17 | 81.37 | 79.24 | 91.69 | 71.13 | 60.45 | 92.90 | 96.61 | 97.14 |
| Variant C | 92.71 | 82.85 | 72.36 | 92.49 | 76.99 | 78.64 | 88.80 | 85.23 | 81.02 | 99.27 | 81.74 | 91.05 | 99.06 | 79.14 | 73.49 | 95.85 | 81.00 | 71.86 | 94.71 | 96.71 | 97.25 |
| Variant D | 92.74 | 82.87 | 72.68 | 92.52 | 77.07 | 78.05 | 88.71 | 85.76 | 81.09 | 99.27 | 82.10 | 91.07 | 99.08 | 81.75 | 79.79 | 95.87 | **82.35** | **72.98** | 94.74 | 97.17 | 97.60 |
| Variant E | 85.80 | 63.98 | 70.09 | 89.09 | 70.85 | 85.42 | 83.25 | 82.18 | 75.06 | 99.00 | **82.21** | 91.02 | 98.92 | 81.21 | 76.62 | 95.12 | 82.11 | 70.04 | 94.18 | 96.97 | 97.50 |
| Variant F | 87.47 | 67.11 | 64.10 | 88.32 | 69.16 | 83.98 | 82.42 | 76.18 | 74.12 | 99.08 | 76.09 | 88.75 | 98.76 | 72.41 | 70.03 | 94.74 | 63.94 | 63.56 | 94.17 | 96.15 | 96.29 |
| DyGMamba | **93.35** | **83.02** | **73.63** | **92.65** | **77.77** | **80.86** | **89.21** | **85.89** | **81.11** | **99.32** | 81.80 | **91.15** | **99.15** | **81.77** | **79.86** | **95.91** | 81.03 | 71.95 | **94.77** | **97.35** | **97.68** |

Table 4: Ablation studies under inductive setting. R/I means random/inductive NSS. Metric is AP.

| Datasets | LastFM | | Enron | | MOOC | | Reddit | | Wikipedia | | UCI | | Social Evo. | |
|---|---|---|---|---|---|---|---|---|---|---|---|---|---|---|
| Models | R | I | R | I | R | I | R | I | R | I | R | I | R | I |
| Variant A | 94.12 | 73.03 | 85.97 | 61.43 | 84.25 | 76.16 | 98.84 | 65.19 | 98.49 | 70.98 | 93.23 | 70.84 | 92.99 | 96.54 |
| Variant B | 94.25 | 75.26 | 89.13 | 67.87 | 86.21 | 75.08 | 97.32 | 58.22 | 92.41 | 70.76 | 90.42 | 60.43 | 91.11 | 96.32 |
| Variant C | 94.18 | 76.44 | 89.40 | 68.33 | 88.59 | 80.39 | 98.90 | 64.07 | 98.65 | 69.82 | 94.47 | 72.05 | 93.07 | 96.20 |
| Variant D | 94.21 | 76.64 | 89.44 | 67.91 | 88.29 | **80.86** | 98.91 | 65.10 | 98.69 | 71.10 | 94.51 | 73.50 | 93.10 | 96.75 |
| Variant E | 88.36 | 55.95 | 89.08 | 68.05 | 82.24 | 74.72 | 98.70 | 65.20 | 98.40 | 70.99 | 93.47 | **74.64** | 92.76 | 96.50 |
| Variant F | 87.71 | 66.45 | 84.02 | 66.85 | 81.13 | 73.85 | 98.54 | 61.74 | 98.37 | 67.59 | 92.81 | 64.59 | 92.42 | 95.38 |
| DyGMamba | **94.42** | **76.76** | **89.67** | **68.77** | **88.64** | 80.75 | **98.97** | **65.30** | **98.77** | **71.14** | **94.76** | 72.17 | **93.13** | **96.83** |

### 4.2.2 Ablation Study

We conduct four ablation studies to study the effectiveness of model components. In study A, we make a model variant (Variant A) by removing the time-level SSM block and restrain our model from learning temporal patterns (information selection is substituted by mean pooling over the output of Eq. 4). In study B, we make a model variant (Variant B) by removing the Mamba SSM layers (Eq. 3) in the node-level SSM block. In study C, we switch the computation of the selection weights $\beta_\theta$ in Eq. 6d to $\beta_\theta = \text{Softmax}(f_{\text{sel}}(\mathbf{H}_\theta^t))$ $(f_{\text{sel}}(\cdot) : \mathbb{R}^{4d} \to \mathbb{R}^{4d})$ to create Variant C. In study D, we base on Variant C and develop Variant D that further enables information selection from opposite nodes, i.e., $\beta_u = \text{Softmax}(f_{\text{sel}}(\mathbf{H}_v^t))/ \beta_v = \text{Softmax}(f_{\text{sel}}(\mathbf{H}_u^t))$. Both ablation C and D do information selection without learning temporal patterns. In study E and F, we devise Variant E and Variant F by removing the time features and the node interaction frequency features, respectively, during neighbor encoding. From Table 3 and 4, we find that: (1) Variant A is constantly beaten by DyGMamba, showing the effectiveness of dynamic information selection based on edge-specific temporal patterns; (2) DyGMamba always outperforms Variant B, indicating the importance of encoding the one-hop temporal neighbors with SSM layers for capturing graph dynamics; (3) Variant C and D perform better than Variant A in most cases, implying that selecting temporal information is generally contributive; (4) Variant C generally lags behind Variant D, meaning that information selection from opposite node is beneficial; (5) DyGMamba performs better than both Variant C and D in almost all cases, proving that information selection based on temporal patterns is more effective; (6) Variant E and F in general achieve worse results than DyGMamba, validating the contribution of both time and node interaction frequency features.

### 4.2.3 A Closer Look into Temporal Pattern Modeling with Synthetic Datasets

We observe from ablation studies that dynamic information selection based on temporal patterns contributes to better model performance on real-world datasets. To better quantify its benefits, we construct three

synthetic datasets, i.e., S1, S2 and S3, that follow different patterns and compare our model with DyGFormer, CTAN as well as Variant A, C, D on them. Each synthetic dataset contains 7 nodes, where the interactions of each pair of two nodes follow a certain pattern along time. And for each node, we generate interactions with all the other nodes. Assume we have a pair of node $u$ and $v$ and they have interactions at $\{t_i\}_{i=0}^N$, in S1, the time intervals between neighboring interactions $\{t_1 - t_0, ..., t_N - t_{N-1}\}$ follow an increasing trend with a constant velocity of 0.05, i.e., $(t_{i+2} - t_{i+1}) - (t_{i+1} - t_i) = 0.05$ . In S2, we set the time intervals to a decreasing trend with the same velocity, i.e., $(t_{i+1} - t_i) - (t_{i+2} - t_{i+1}) = 0.05$. And in S3, we modify S1 by repeating several periods of increasing patterns taken from S1 to form a periodic dataset. In this way, we have three datasets demonstrating diverse temporal patterns: increasing/decreasing/periodic time intervals between neighboring interactions. Details of dataset construction and statistics are provided in App. A.2. From Table 5, we observe that DyGMamba greatly outperforms DyGFormer and CTAN. More importantly, Variant A, C and D show similar performance to DyGFormer, meaning that our time-level SSM block is able to capture temporal patterns and modeling such patterns for dynamic information selection is important in CTDG reasoning. For more implementation details on synthetic datasets, please refer to App. C.2.

Table 5: Performance (Random NSS) on synthetic datasets.

(a) AP on synthetic datasets.

| Datasets | DyGFormer | CTAN | Variant A | Variant C | Variant D | DyGMamba |
|---|---|---|---|---|---|---|
| S1 | $55.19 \pm 0.98$ | $51.25 \pm 2.11$ | $53.72 \pm 0.04$ | $55.45 \pm 0.32$ | $54.52 \pm 0.71$ | $\mathbf{81.58 \pm 1.31}$ |
| S2 | $57.80 \pm 4.61$ | $51.17 \pm 0.93$ | $60.16 \pm 2.20$ | $64.71 \pm 2.33$ | $61.51 \pm 3.00$ | $\mathbf{85.36 \pm 2.55}$ |
| S3 | $79.20 \pm 0.60$ | $51.46 \pm 0.19$ | $77.61 \pm 2.31$ | $77.61 \pm 2.31$ | $79.41 \pm 2.13$ | $\mathbf{86.59 \pm 0.09}$ |

(b) AUC-ROC on synthetic datasets.

| Datasets | DyGFormer | CTAN | Variant A | Variant C | Variant D | DyGMamba |
|---|---|---|---|---|---|---|
| S1 | $56.27 \pm 0.54$ | $51.25 \pm 2.38$ | $53.16 \pm 0.39$ | $57.41 \pm 0.04$ | $55.83 \pm 1.03$ | $\mathbf{86.61 \pm 1.30}$ |
| S2 | $59.06 \pm 6.07$ | $51.46 \pm 0.91$ | $62.50 \pm 2.28$ | $64.93 \pm 2.59$ | $62.06 \pm 3.40$ | $\mathbf{89.94 \pm 2.70}$ |
| S3 | $82.89 \pm 1.34$ | $52.12 \pm 0.44$ | $81.78 \pm 0.40$ | $82.73 \pm 1.97$ | $84.20 \pm 3.57$ | $\mathbf{91.72 \pm 0.11}$ |

## 4.3 Efficiency Analysis

We evaluate the models' efficiency based on the following aspects: model size (number of trainable parameters), per-epoch training time, and GPU memory consumption during training. Since DyGMamba shares similarities with DyGFormer, i.e., both of them model large sequences of one-hop temporal neighbors and use patching to enhance scalability, we further analyze the impact of patch size on their scalability and performance. We also compare the complexity of DyGMamba and DyGFormer to highlight DyGMamba's efficiency.

### 4.3.1 Model Size, Per Epoch Training Time and GPU Memory Comparison Across Various Models

Fig. 2 compares DyGMamba with five baselines in terms of number of parameters (model size), per epoch training time, and GPU memory consumption during training[5]. We find that: (1) DyGMamba uses very few parameters while maintaining the best performance, showing a strong parameter efficiency. Only CTAN constantly uses fewer parameters than DyGMamba, however, its performance is significantly worse with the only exception of Enron; (2) DyGMamba is always more efficient than DyGFormer with the same length of input sequence ($\rho/p =$32) while achieving the same performance; (3) Although DyGMamba generally consumes more GPU memory and takes more time to train per epoch compared with most baselines, the gap of consumption is modest. To model more temporal neighbors for long-range temporal dependent datasets, DyGMamba naturally requires more computational resources, thus enlarging the consumption gap. DyGFormer shows the same trend as DyGMamba since it also captures long-term temporal dependencies but at a higher cost; (4) CTAN requires very few computational resources. However, on long-range temporal dependent datasets, it is beaten by DyGFormer and DyGMamba by a large margin, e.g., on LastFM and

---

[5]The baselines not included here are either extremely inefficient (e.g., CAWN) or inferior in performance (e.g., DyRep). Complete statistics of all baseline models presented in App. G.1

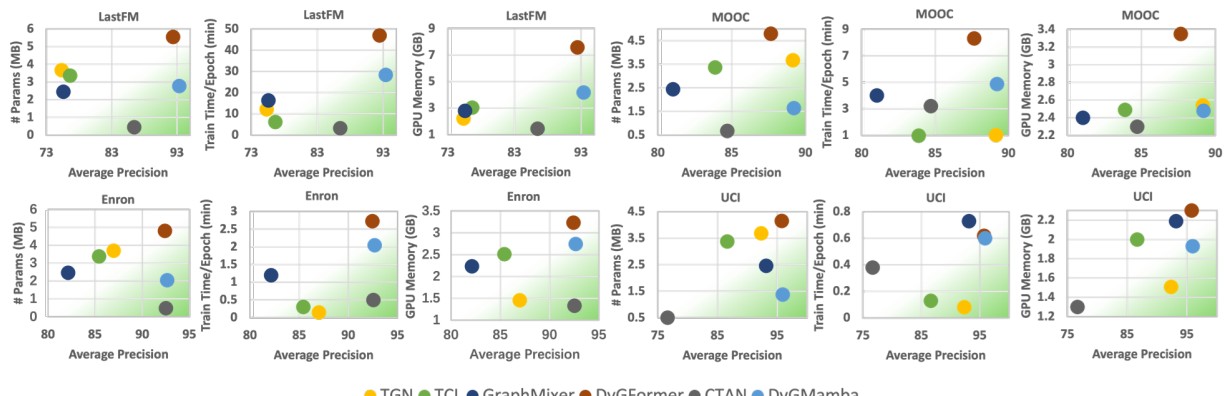

Figure 2: Efficiency comparison on four datasets among DyGMamba and five baselines in terms of number (#) of parameters, training time per epoch and GPU memory. The performance metric here is AP of transductive link prediction under random NSS. The greener, the better performance/efficiency. In contrast to other methods, DyGMamba consistently shows strong overall capability across different datasets. More explanations in Sec. 4.3.1. Further comparison on Reddit and Social Evo. is presented in App. G.1

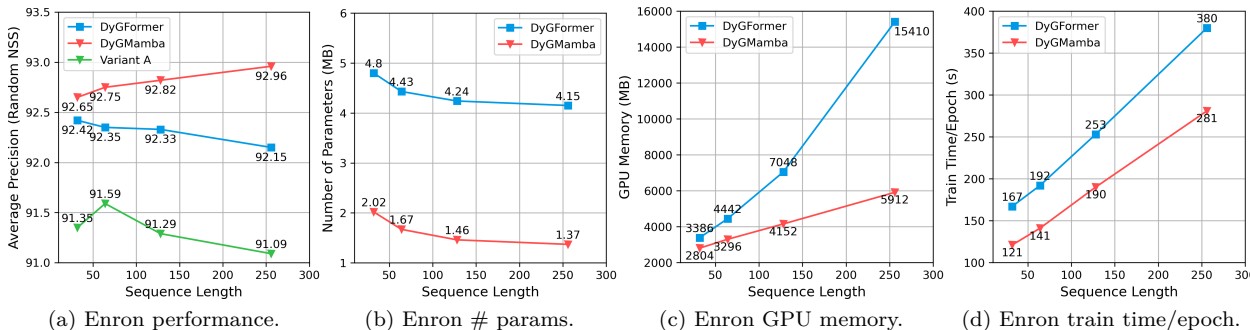

Figure 3: Impact of patch size on DyGFormer, DyGMamba and Variant A, given a fixed number of sampled temporal neighbors $\rho$ on Enron. Patch size $p$ varies from 8, 4, 2, 1. Sequence length $\rho/p$ increases as patch size decreases. Performance is the transductive AP under random NSS.

Enron. Besides, CTAN is also hard to converge. Although it takes little time to train a single epoch, it needs more epochs to reach the best performance, leading to a long total training time. See App. G.2 for a total training time comparison among DyGFormer, CTAN and DyGMamba. To supplement, we also present in App. G.3 another experiment to compare DyGMamba with baselines in modeling an increasing number of temporal neighbors on Enron with limited total training time.

### 4.3.2 Impact of Patch Size on Scalability and Performance

Patching treats $p$ temporal neighbors as one patch and thus decreases the sequence length by $p$ times. This is very helpful in cutting the consumption of GPU memory and training/evaluation time. However, patching introduces excessive parameters because it is done through $f_N$, $f_E$, $f_T$ and $f_F$ whose sizes increase as the patch size grows. Fig. 3b shows the numbers of parameters of DyGFormer and DyGmamba with different patch sizes on a long-term temporal dependent dataset Enron. We find that patching greatly affects model sizes. To further study how patching affects DyGMamba, we decrease the patch size gradually from 8 to 1 and track DyGMamba's performance (Fig. 3a) as well as efficiency (Fig. 3b to 3d) on Enron. Meanwhile, we also keep track on DyGFormer under the same patch size for comparison. We have several findings: (1) Whatever the patch size is, DyGMamba always consumes fewer parameters, less GPU memory and per epoch training time, showing its high efficiency; (2) While both models require increasing computational budgets as the patch size decreases, the speed of increase is much lower for DyGMamba, demonstrating its strong scalability in modeling longer sequences; (3) Different trends in performance change are observed between two models. While DyGFormer performs worse, DyGMamba can benefit from a smaller patch size,

indicating its strong ability to capture nuanced temporal details even if the sequence becomes much longer. Note that the models use fewer parameters under smaller patch size. This also shows that DyGMamba can achieve much stronger parameter efficiency by reducing patch sizes. To further study the reason for finding (3), we plot the performance of Variant A under different patch sizes in Fig. 3a. We find that Variant A's performance degrades when sequence length is more than 64. This means that dynamic information selection based on edge-specific temporal patterns is essential for DyGMamba to optimally process long sequences. To supplement, we provide additional analysis on MOOC in App. J.

**Reason for Diverse Performance Trends between DyGMamba and DyGFormer.** Patching decreases sequence length by applying linear transformation on a patch of node embeddings, giving a model much more parameters to tune (as indicated in Fig. 3b). Larger patch size mixes more sampled neighbors in each patch, introducing more training parameters while losing nuanced temporal details brought by the temporal order of the neighbors within each patch. For DyGFormer, the negative influence brought by mixing neighbors within patches is smaller than the positive influence brought by more trainable parameters, so the performance constantly increases with a growing patch size in Fig. 3a. By contrast, for DyGMamba, the negative influence brought by mixing neighbors is much greater than the positive influence brought by more trainable parameters, so it shows better performance when the patch size is smaller. For Variant A, given an increasing patch size, it follows the trend of DyGFormer when the patch size is below a threshold and shows degrading performance after that. This means that there is a trade-off between the lost temporal details and the additional parameters when we modify patch size. Also, by comparing Variant A and DyG-Mamba, we can tell that the difference in performance trend roots from the dynamic information selection module. Smaller patch size leads to longer sequences with more temporal details. The strong capability of the dynamic information selection module in long-range temporal reasoning let DyGMamba benefit from more nuanced temporal details, which is more influential than increasing training parameters.

### 4.3.3 Complexity Analysis: DyGMamba vs. DyGFormer

DyGMamba follows the current state-of-the-art DyGFormer by learning from one-hop temporal neighbors for temporal reasoning. We analyze the complexity of both models to show DyGMamba's efficiency. Sequence length is the key factor affecting the consumption of computational resources in DyGFormer and DyGMamba. Following the computation of previous work (Zhu et al., 2024), the complexity of Transformer and Mamba in DyGFormer and DyGMamba can be written as

$$O(\text{Transformer}) = 4(\rho/p)(4d)^2 + 2(\rho/p)^2(4d) = 64(\rho/p)d^2 + 8d(\rho/p)^2, \tag{8a}$$

$$O(\text{Mamba}) = 3(\rho/p)(4d)d_{\text{SSM}} + (\rho/p)(4d)d_{\text{SSM}} = 16d_{\text{SSM}}d(\rho/p). \tag{8b}$$

This means that DyGMamba holds a computational complexity linear to $\rho/p$, while DyGFormer's complexity is quadratic to $\rho/p$. As a result, as the sequence length grows (either $\rho$ increases or $p$ decreases), DyGFormer is less scalable compared with DyGMamba[6]. Some may argue that increasing the patch size $p$ to a large enough value can offset the negative influence of the higher complexity of Transformer. However, as discussed in Sec. 4.3.2, increasing patch size will substantially increase model parameters, causing burden in parameter optimization, and meanwhile lose temporal details. This indicates that patching is not always beneficial and DyGMamba's low complexity provides an alternative way to maintain great efficiency while considering more temporal information. With the same number of sampled temporal neighbors and equivalent computational resources, DyGMamba can leverage a smaller patch size, mitigating the negative effects of lost temporal details and simplifying parameter optimization, as implied in Sec. 4.3.2.

**Modeling Increasing Number of Temporal Neighbors with a Fixed Patch Size.** Sequence length is decided by the sampled temporal neighbors $\rho$ and the patch size $p$. If we want to model a huge number of temporal neighbors, e.g., $\rho = 4096$, keeping the sequence length unchanged as 32 means we need to set $p$ to 128. Considering the potential negative impact brought by excessively large patch size (i.e., too many lost temporal details, harder optimization and much lower parameter efficiency), it is also important to see

---

[6]As we set $k$ (number of edge-specific historical interactions discussed in Sec. 3.2) to a number much smaller than the number of sampled one-hop temporal neighbors, i.e., sequence length $\rho/p$, we omit here the contribution of the time-level SSM in complexity analysis.

if a model is scalable to $\rho$, with a fixed $p$. In Fig. 4, we show the consumed GPU memory of DyGMamba and DyGFormer with $\rho$ varying from 64, 128, 256, 512, 1024, 2048, 4096 and 8192 on Enron, under a fixed patch size $p = 8$. We find that DyGMamba shows superior scalability over DyGFormer. While DyGFormer can only process 2048 nodes, DyGMamba can deal with more than 8192 (at least 4 times) on a single 45GB NVIDIA A40 GPU. This implies our model's potential to process a huge number of temporal neighbors with a limited GPU memory budget, while keeping a moderate patch size. The experiment provides a concrete example to demonstrate DyGMamba's advantage in efficiency brought by its low complexity.

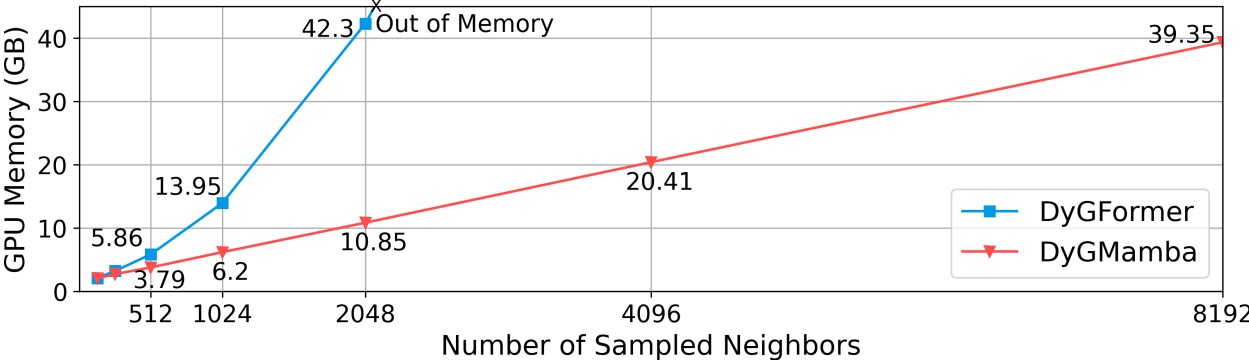

Figure 4: GPU memory consumption on Enron with increasing neighbors. For clarity, memory is shown only when neighbors are more than 512.

## 5 Discussion

In this section, we discuss where DyGMamba excels and where it falls short, and explain our choice to consider only one-hop temporal neighbors. We additionally provide a full justification of our motivation in App. L, recommended for reading after the main paper.

**Where DyGMamba Excels and Where It Falls Short.** We summarize here the conditions under which DyGMamba performs well and scenarios where it may fall short. DyGMamba is expected to outperform previous state-of-the-art models in the following scenarios: (1) when modeling long-range temporal information is critical; (2) when there is a clear temporal pattern, i.e., the density of repeated edges follows a specific trend. For the first scenario, we have demonstrated in Table 1 and 2 that DyGMamba surpasses previous models on long-range temporal-dependent datasets (LastFM, MOOC, and Enron) in both transductive and inductive link prediction under the random NSS setting. Under the historical and inductive NSS settings, DyGMamba consistently ranks among the top 3 models and even achieves the top 1 position in 7 out of 9 cases. Regarding the second scenario, Table 5 confirms that DyGMamba's dynamic information selection module significantly enhances its capability to model temporal patterns, enabling superior performance on datasets characterized by clear temporal trends. Conversely, DyGMamba might underperform previous state-of-the-art models on certain datasets (e.g., Wikipedia and UCI) under historical and inductive NSS settings, particularly when the number of available one-hop temporal neighbors for nodes in the link prediction query is limited (the availability of historical neighbors across datasets for both link prediction tasks under all NSS settings is presented in Table 22). DyGMamba rely on abundant historical information to effectively capture graph dynamics and temporal patterns. Insufficient historical context may hinder its performance, leading to potential misclassification of negative edges and higher false positive rates.

**Why We Only Consider One-Hop Neighbors.** We only consider encoding one-hop temporal neighbors in DyGMamba due to two reasons. First, our primary motivation is to design an efficient model that effectively captures long-range temporal dependencies in CTDGs. While incorporating higher-order information is an intuitive approach, it introduces several challenges. Maintaining the same computational budget would require reducing the number of first-order (one-hop) temporal neighbors, leading to a less representative one-hop neighborhood. The alternative—allocating more computational resources such as increased GPU memory—goes against our objective of efficiency in this work. Moreover, modeling higher-order temporal

neighbors requires additional specialized modules to differentiate the influence of neighbors from different hops, which can further increase computational cost and difficulties in modeling. Besides, recent CTDG models that only consider one-hop temporal neighbors, such as DyGFormer and FreeDyG, have demonstrated strong performance, and DyGMamba further validates this observation. Some of our baselines, such as CAWN and CTAN, do incorporate higher-order information. However, as shown in Table 1 and 2, they are outperformed by DyGMamba in most cases. A potential reason for this is that these models may still struggle to optimally differentiate information from different hops. Efficiently integrating higher-order information into CTDG modeling is a promising research direction, and we would be happy to explore it in future work.

## 6 Conclusion

We propose DyGMamba, an efficient CTDG representation learning model that can capture long-term temporal dependencies. DyGMamba first leverages a node-level SSM to encode long sequences of historical node interactions. It then employs a time-level SSM to learn edge-specific temporal patterns. The learned patterns are used to select the critical part of the encoded temporal information. DyGMamba achieves superior performance on dynamic link prediction, and moreover, it shows high efficiency and strong scalability compared with previous CTDG methods, implying a great potential in modeling huge amounts of temporal information with a limited computational budget.

## 7 Limitation

One limitation of our work is that DyGMamba is designed to only model CTDGs. DTDGs are represented as a sequence of graph snapshots, where all the edges in a snapshot are taken as existing simultaneously. This poses a challenge to DyGMamba because it can only encode edges sequentially, which are not suitable for modeling concurrent edges. A possible solution is to first employ methods such as graph neural networks (GNN) to encode each graph snapshot and then use SSM to model the temporal graph dynamics (similar to approaches used in recent works on temporal knowledge graph modeling, e.g., Han et al. (2021)). However, the introduction of GNNs inevitably requires more computational resources, which would lower model efficiency.

## Acknowledgements

Zifeng Ding receives funding from the European Research Council (ERC) under the European Union's Horizon 2020 Research and Innovation programme grant AVeriTeC (Grant agreement No. 865958). He also acknowledges travel support from the European Union's Horizon 2020 research and innovation programme under ELISE Grant Agreement No 951847. Yuan He is supported by Samsung Research UK (SRUK), EPSRC projects UK FIRES (EP/S019111/1) and ConCur (EP/V050869/1). Michael Bronstein and Antonio Norelli are supported by EPSRC Turing AI World-Leading Research Fellowship No. EP/X040062/1, EPSRC AI Hub on Mathematical Foundations of Intelligence: An "Erlangen Programme" for AI No. EP/Y028872/1. Antonio Norelli is also supported by Project CETI. Jingcheng Wu is funded by the Deutsche Forschungsgemeinschaft (DFG, German Research Foundation) - SFB 1574 - Project number 471687386. This work is partially supported by the Munich Center for Machine Learning and is performed on the HoreKa supercomputer funded by the Ministry of Science, Research and the Arts Baden-Württemberg and by the Federal Ministry of Education and Research. The authors also gratefully acknowledge the Gauss Centre for Supercomputing e.V. for funding this project by providing computing time on the GCS Supercomputer HLRS at High-Performance Computing Center Stuttgart.

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

## A    CTDG Dataset Details

### A.1    Real-World Benchmark Datasets

We present the dataset statistics of all considered CTDG datasets in Table 6. All the datasets in our experiments are taken from Yu et al. (2023). We chronologically split each dataset with the ratio of 70%/15%/15% for training/validation/testing. Please refer to it for detailed dataset descriptions.

Table 6: Dataset statistics. # N&E Feat means the numbers of node and edge features.

| Datasets | # Nodes | # Edges | # N&E Feat | Bipartite | Duration | # Timestamps | Time Granularity |
|---|---|---|---|---|---|---|---|
| LastFM | 1,980 | 1,293,103 | 0 & 0 | True | 1 month | 1,283,614 | Unix timestamps |
| Enron | 184 | 125,235 | 0 & 0 | False | 3 years | 22,632 | Unix timestamps |
| MOOC | 7,144 | 411,749 | 0 & 4 | True | 17 months | 345,600 | Unix timestamps |
| Reddit | 10,984 | 672,447 | 0 & 172 | True | 1 month | 669,065 | Unix timestamps |
| Wikipedia | 9,227 | 157,474 | 0 & 172 | True | 1 month | 152,757 | Unix timestamps |
| UCI | 1,899 | 59,835 | 0 & 0 | False | 196 days | 58,911 | Unix timestamps |
| Social Evo. | 74 | 2,099,519 | 0 & 2 | False | 8 months | 565,932 | Unix timestamps |

## A.2 Synthetic Datasets

For all of our three synthetic datasets, node and link features are not involved during dataset construction. The construction details are as follows:

- **S1**: For each interaction pair $u$ and $v$, their first interaction is at timestamp 0 and the second interaction is generated randomly. Thus, the first time interval is also determined. Starting from the second interval, they follow an increasing trend with a constant velocity of 0.05, i.e., $(t_{i+2} - t_{i+1}) - (t_{i+1} - t_i) = 0.05$. The number of interactions for each node pair is randomly determined and the interaction numbers of all node pairs sum up to 100000.

- **S2**: For each interaction pair $u$ and $v$, their first interaction is at timestamp 0 and the second interaction is generated randomly. However, it should be large enough so that the interval will not drop to zero or negative afterwards. Starting from the second interval, they follow a decreasing trend with a constant velocity of 0.05, i.e., $(t_{i+1} - t_i) - (t_{i+2} - t_{i+1}) = 0.05$. The number of interactions for each node pair is randomly determined and the interaction numbers of all node pairs sum up to 100000.

- **S3**: S3 contains 8 periods. In each period, the interactions of each node pair $u$ and $v$ are generated following the same pattern in S1. The number of interactions for each node pair is randomly determined and the interaction numbers of all node pairs sum up to 12000 in the period.

We present the statistics of all synthetic datasets in Table 7. We chronologically split each dataset with the ratio of 70%/15%/15% for training/validation/testing. To better visualize the temporal patterns in each dataset, we pick one pair of interacting nodes and plot the time intervals between neighboring interactions in each dataset in Figure 5. Note that for the periodic dataset S3 (Figure 5c), each of the train, validation and test sets contains at least one start of a new period. This ensures that models have to capture periodic temporal patterns in order to achieve good performance during evaluation, rather than only learning the increasing time intervals as specified in S1.

Furthermore, we provide the information about the numbers of interactions regarding interacting node pairs in Table 8. We show that each node pair is equipped with a substantial number of interactions, meaning that temporal patterns in our synthetic datasets span across long time periods. This encourages models to consider long-term temporal dependencies for better graph reasoning.

Table 7: Synthetic dataset statistics.

| Datasets | # Nodes | # Edges | # Timestamps | Time Range |
|---|---|---|---|---|
| S1 | 7 | 100,000 | 96,869 | 0 - 163241.65 |
| S2 | 7 | 100,000 | 98,004 | 0 - 1573561.52 |
| S3 | 7 | 95,657 | 95,370 | 0 - 1771300.40 |

Table 8: Interaction information of node pairs in synthetic datasets. Complete Dataset includes the numbers of interactions across the whole datasets, including training, validation and testing.

| | Datasets | Avg. # Interactions | Min # Interactions | Max # Interactions |
|---|---|---|---|---|
| Training Set | S1 | 1,428.57 | 1,205 | 1,663 |
| | S2 | 1,428.57 | 1,424 | 1,433 |
| | S3 | 1,367.91 | 1,364 | 1,372 |
| Complete Dataset | S1 | 2,010.20 | 1,879 | 2,150 |
| | S2 | 2,010.20 | 1,921 | 2,097 |
| | S3 | 1,952.18 | 1,721 | 2,193 |
| | S3 (each period) | 244.02 | 215 | 274 |

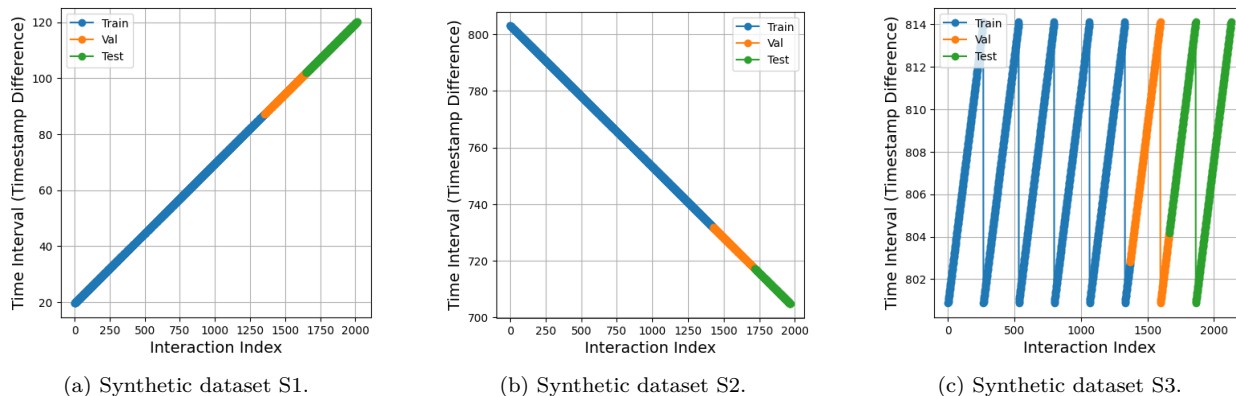

(a) Synthetic dataset S1.  (b) Synthetic dataset S2.  (c) Synthetic dataset S3.

Figure 5: Time intervals of a node pair $u$, $v$ in synthetic datasets S1, S2 and S3.

# B  Baseline Details

We provide the detailed descriptions of all baselines here. The baselines can be split into two groups: the methods designed/not designed for long-range temporal information propagation.

## B.1  Baselines Not Designed for Long-Range Temporal Information Propagation

- **JODIE** (Kumar et al., 2019): JODIE employs a recurrent neural network (RNN) for each node and uses a projection operation to learn the future representation trajectory of each node.

- **DyRep** (Trivedi et al., 2019): DyRep updates node representations as events appear. It designs a two-time scale deep temporal point process approach for source and destination nodes and couples the structural and temporal components with a temporal-attentive aggregation module.

- **TGAT** (Xu et al., 2020): TGAT computes the node representations by aggregating each node's temporal neighbors based on a self-attention module. A time encoding function is proposed to learn functional representations of time.

- **TGN** (Rossi et al., 2020): TGN leverages an evolving memory for each node and updates the memory when a node-relevant interaction occurs by using a message function, a message aggregator, and a memory updater. An embedding module is used to generate the temporal representations of nodes.

- **CAWN** (Wang et al., 2021c): CAWN is a random walk-based method. It does multiple causal anonymous walks for each node and extracts relative node identities from the walk results. RNNs are then introduced to encode the anonymous walks. The aggregated walk information forms the final node representation.

- **EdgeBank** (Poursafaei et al., 2022): EdgeBank is a non-parametric method purely based on memory. It stores the observed interactions in its memory and updates the memory through various strategies. An interaction, i.e., link, will be predicted as existing if it is stored in the memory, and non-existing otherwise. EdgeBank uses four memory update strategies: (1) $EdgeBank_\infty$, where all the observed edges are stored in the memory; (2) $EdgeBank_{tw-ts}$, where only the edges within the duration of the test set from the immediate past are kept in the memory; (3) $EdgeBank_{tw-re}$, where only the edges within the average time intervals of repeated edges from the immediate past are kept in the memory; (4) $EdgeBank_{th}$, where the edges with appearing counts higher than a threshold are stored in the memory. The results reported in our paper correspond to the best results achieved among the four memory update strategies.

- **TCL** (Wang et al., 2021a): TCL first extracts temporal dependency interaction sub-graphs for source and interaction nodes and then presents a graph transformer to aggregate node information from the sub-graphs. A cross-attention operation is implemented to enable information communication between two source and destination nodes.

- **GraphMixer** (Cong et al., 2023): GraphMixer designs a link-encoder based on MLP-Mixer (Tolstikhin et al., 2021) to learn from the temporal interactions. A mean pooling-based node-encoder is used to aggregate the node features. Link prediction is done with a link classifier that leverages the representations output by link-encoder and node-encoder.

Note that TGN uses a memory network to store the whole graph history, making it able to preserve long-range temporal information. However, as discussed in Yu et al. (2023), it faces a problem of vanishing/exploding gradients, preventing it from optimally capturing long-term temporal dependencies. EdgeBank can also preserve a very long graph history, but we can observe from the experimental results (Table 1, 12) that without learnable parameters, it is not strong enough on long-range temporal dependent datasets.

### B.2 Baselines Designed for Long-Range Temporal Information Propagation

- **DyGFormer** (Yu et al., 2023): DyGFormer is a Transformer-based CTDG model. It takes the long-term one-hop temporal interactions of source and destination nodes and uses a Transformer to encode them. A patching technique is developed to cut the computational consumption and a node co-occurrence encoding scheme is used to exploit the correlations of nodes in each interaction. DyGFormer achieves long-range temporal information propagation by increasing the number of sampled one-hop historical interactions. The patching technique ensures that even with a huge number of sampled interactions, the length of the sequence input into Transformer will not be too long, making it possible to implement DyGFormer with a limited computational budget.

- **CTAN** (Gravina et al., 2024): CTAN is deep graph network for learning CTDGs based on non-dissipative ordinary differential equations. CTAN's formulation allows for a scalable long-range temporal information propagation in CTDGs because its non-dissipative layer can retain the information from a specific event indefinitely, ensuring that the historical context of a node is preserved despite the occurrence of additional events involving this node.

## C  Implementation Details

We train every CTDG model except for CTAN for a maximum number of 200 epochs. Maximum epochs for CTAN training is 1000. We evaluate each model on the validation set at the end of every training epoch and adopt an early stopping strategy with a patience of 20. We take the model that achieves the best validation result for testing. We use the implementations[7] provided by Yu et al. (2023) for all baseline models except CTAN. For CTAN, we use its official implementation[8]. All models are trained with a batch size of 200 for fair efficiency analysis. All experiments are implemented with PyTorch (Paszke et al., 2019) on a server

---

[7]https://github.com/yule-BUAA/DyGLib
[8]https://github.com/gravins/non-dissipative-propagation-CTDGs

equipped with an AMD EPYC 7513 32-Core Processor and a single NVIDIA A40 with 45GB memory. We run each experiment for five times with five random seeds and report the mean results together with error bars.

## C.1 Hyperparameter Configurations on Real-World Datasets

For all the baselines except CTAN, please refer to Yu et al. (2023) for the hyperparameter configurations on real-world datasets. For CTAN, we present its hyperparameter configurations in Table 9. We keep its hyperparameters unchanged for all real-world datasets. Note that we set the number of graph convolution layers (GCLs) in CTAN to its maximum, i.e., 5, in order to maximize its performance in capturing long-term temporal dependencies.

Table 9: Hyperparameter configurations of CTAN on all real-world datasets. $\gamma$ here denotes the discretization step size introduced in Gravina et al. (2024), different from the one in DyGMamba.

| Model | # GCL | $\epsilon$ | $\gamma$ | Embedding Dim |
|-------|-------|-----------|----------|---------------|
| CTAN  | 5     | 0.5       | 0.5      | 128           |

We report the hyperparameter searching strategy of DyGMamba on real-world datasets and the best hyperparameters in Table 10. To achieve fair efficiency comparison with DyGFormer, we fix the length of the input sequence into the node-level SSM to 32, i.e., $\rho$ & $p = 32$. The results reported in Table 1, 2, 12, 13 are all achieved by DyGMamba with $\rho$ & $p = 32$. In practice, we can decrease $p$ to have a better performance given more computational resources (as discussed in Sec. 4.3). DyGMamba keeps the embedding size as same as DyGFormer on all real-world datasets, i.e., $d_N = d_E = 172$, $d_T = 100$, $d_F = 50$. We also set $\gamma = 0.5$ for all experiments of DyGMamba. The dimension of SSMs $d_{SSM} = 16$ remains the default value of mamba SSM's official repository[9]. For all experiments, we set the numbers of layers in both node-level and time-level SSMs as 2. We also set $\gamma$ to 0.5 for all datasets. A smaller $\gamma$ lowers the computational resource consumption in time-level SSM, potentially at the cost of performance. Raising $\gamma$'s value does not necessarily lead to better performance but will lower efficiency. We search $\gamma$'s value in {0.1, 0.5, 0.7, 1} and find that 0.5 brings a good balance between performance and efficiency.

Table 10: DyGMamba hyperparameter searching strategy on real-world datasets. The best settings are marked as bold.

| Datasets | Dropout | $\rho$ & $p$ | $k$ |
|----------|---------|--------------|-----|
| LastFM | {0.0, **0.1**, 0.2} | {1024 & 32, **512 & 16**, 256 & 8} | {30, **10**, 5} |
| Enron | {**0.0**, 0.1, 0.2} | {512 & 16, **256 & 8**, 128 & 4} | {**30**, 10, 5} |
| MOOC | {0.0, **0.1**, 0.2} | {512 & 16, 256 & 8, **128 & 4**} | {30, **10**, 5} |
| Reddit | {0.0, 0.1, **0.2**} | {128 & 4, **64 & 2**, 32 & 1} | {30, 10, **5**} |
| Wikipedia | {0.0, **0.1**, 0.2} | {64 & 2, **32 & 1**} | {30, 10, **5**} |
| UCI | {0.0, **0.1**, 0.2} | {64 & 2, **32 & 1**} | {30, 10, **5**} |
| Social Evo. | {0.0, **0.1**, 0.2} | {64 & 2, **32 & 1**} | {30, 10, **5**} |

## C.2 Hyperparameter Configurations on Synthetic Datasets

We use the same settings of CTAN on real-world datasets when we experiment it on synthetic datasets. For DyGFormer and DyGMamba, we fix the length of the input sequence into the Transformer and the node-level SSM to 32, i.e., $\rho/p = 32$. For DyGFormer, we set the hyperparameters except $\rho$ and $p$ to the same default values as on real-world datasets, and search for the best $\rho$ & $p$ within {512 & 16, 256 & 8, 128 & 4, 64 & 2, 32 & 1}. For DyGMamba, we search for the best $\rho$ & $p$ within the same search range and further search for the best $k$. All the other hyperparameters are set as same as the setting on LastFM. We

---

[9]https://github.com/state-spaces/mamba

Table 11: DyGFormer and DyGMamba hyperparameter searching strategy on synthetic datasets. The best settings are marked as bold.

| Models | DyGFormer | DyGMamba | |
|---|---|---|---|
| Datasets | $\rho$ & $p$ | $\rho$ & $p$ | $k$ |
| S1 | {512 & 16, 256 & 8, 128 & 4, **64 & 2**, 32 & 1} | {512 & 16, **256 & 8**, 128 & 4, 64 & 2, 32 & 1} | {30, **10**, 5} |
| S2 | {**512 & 16**, 256 & 8, 128 & 4, 64 & 2, 32 & 1} | {**512 & 16**, 256 & 8, 128 & 4, 64 & 2, 32 & 1} | {**30**, 10, 5} |
| S3 | {512 & 16, 256 & 8, 128 & 4, 64 & 2, **32 & 1**} | {**512 & 16**, 256 & 8, 128 & 4, 64 & 2, 32 & 1} | {30, **10**, 5} |

Table 12: AUC-ROC of transductive dynamic link prediction.

| NSS | Datasets | JODIE | DyRep | TGAT | TGN | CAWN | EdgeBank | TCL | GraphMixer | DyGFormer | CTAN | DyGMamba |
|---|---|---|---|---|---|---|---|---|---|---|---|---|
| Random | LastFM | 70.89 ± 1.97 | 71.40 ± 2.12 | 71.47 ± 0.14 | 76.64 ± 4.66 | 85.92 ± 0.16 | 83.77 ± 0.00 | 71.09 ± 1.48 | 73.51 ± 0.14 | 93.03 ± 0.11 | 85.12 ± 0.77 | **93.31 ± 0.18** |
| | Enron | 87.77 ± 2.43 | 83.09 ± 2.20 | 68.57 ± 1.46 | 88.72 ± 0.95 | 90.34 ± 0.23 | 87.05 ± 0.00 | 83.33 ± 0.93 | 84.16 ± 0.34 | 93.20 ± 0.12 | 87.09 ± 1.51 | **93.34 ± 0.23** |
| | MOOC | 84.50 ± 0.60 | 84.50 ± 0.87 | 87.01 ± 0.16 | **91.91 ± 0.82** | 80.48 ± 0.41 | 60.86 ± 0.00 | 84.02 ± 0.59 | 84.04 ± 0.12 | 88.08 ± 0.50 | 85.40 ± 2.67 | 89.58 ± 0.12 |
| | Reddit | 98.29 ± 0.05 | 98.13 ± 0.04 | 98.50 ± 0.01 | 98.61 ± 0.05 | 99.02 ± 0.00 | 95.37 ± 0.00 | 97.67 ± 0.01 | 97.17 ± 0.02 | 99.15 ± 0.01 | 97.24 ± 0.75 | **99.27 ± 0.01** |
| | Wikipedia | 96.36 ± 0.14 | 94.43 ± 0.32 | 96.60 ± 0.07 | 98.37 ± 0.10 | 98.54 ± 0.01 | 90.78 ± 0.00 | 97.27 ± 0.06 | 96.89 ± 0.04 | 98.92 ± 0.03 | 97.00 ± 0.21 | **99.08 ± 0.02** |
| | UCI | 90.35 ± 0.51 | 69.46 ± 2.66 | 78.76 ± 1.10 | 92.03 ± 0.69 | 93.81 ± 0.23 | 77.30 ± 0.00 | 85.49 ± 0.82 | 91.62 ± 0.52 | 94.45 ± 0.22 | 76.25 ± 2.83 | **94.77 ± 0.18** |
| | Social Evo. | 92.13 ± 0.20 | 90.37 ± 0.52 | 94.93 ± 0.06 | 95.31 ± 0.27 | 87.34 ± 0.10 | 81.60 ± 0.00 | 95.45 ± 0.21 | 95.21 ± 0.07 | 96.25 ± 0.04 | Timeout | **96.38 ± 0.02** |
| | **Avg. Rank** | 7.14 | 8.86 | 7.14 | 3.86 | 4.86 | 9.14 | 7.29 | 7.14 | 2.14 | | **1.14** |
| Historical | LastFM | 75.65 ± 4.43 | 70.63 ± 2.56 | 64.23 ± 0.45 | 78.00 ± 2.97 | 67.92 ± 0.32 | 78.09 ± 0.00 | 60.53 ± 2.54 | 64.06 ± 0.34 | 78.80 ± 0.02 | 79.50 ± 0.82 | **79.82 ± 0.27** |
| | Enron | 75.21 ± 1.27 | 76.36 ± 1.42 | 62.36 ± 1.07 | 76.75 ± 1.40 | 65.62 ± 0.49 | 79.59 ± 0.00 | 71.72 ± 1.24 | 74.82 ± 2.04 | 77.35 ± 0.64 | **81.95 ± 1.64** | 77.73 ± 0.61 |
| | MOOC | 82.38 ± 1.75 | 80.71 ± 2.08 | 81.53 ± 0.79 | 86.59 ± 2.03 | 71.74 ± 0.88 | 61.90 ± 0.00 | 73.22 ± 1.21 | 77.09 ± 0.83 | 87.26 ± 0.83 | 73.87 ± 2.77 | **87.91 ± 0.93** |
| | Reddit | 80.70 ± 0.20 | 79.96 ± 0.23 | 82.83 ± 0.27 | 81.04 ± 0.23 | 80.42 ± 0.20 | 78.58 ± 0.00 | 76.83 ± 0.12 | 77.83 ± 0.33 | 80.61 ± 0.48 | **90.63 ± 0.07** | 78.99 ± 1.24 |
| | Wikipedia | 80.71 ± 0.64 | 77.49 ± 0.72 | 82.83 ± 0.27 | 83.28 ± 0.26 | 65.74 ± 3.46 | 77.27 ± 0.00 | 85.55 ± 0.47 | 87.47 ± 0.20 | 72.78 ± 6.65 | **95.43 ± 0.07** | 78.99 ± 1.24 |
| | UCI | 78.21 ± 3.18 | 58.65 ± 3.58 | 57.12 ± 0.98 | **78.48 ± 1.79** | 57.67 ± 1.11 | 69.56 ± 0.00 | 65.42 ± 2.62 | | 77.46 ± 1.63 | 75.71 ± 0.57 | 75.43 ± 1.99 |
| | Social Evo. | 91.83 ± 1.52 | 92.81 ± 0.60 | 93.63 ± 0.48 | 94.27 ± 1.33 | 87.61 ± 0.06 | 85.81 ± 0.00 | 95.03 ± 0.82 | 94.65 ± 0.28 | 97.16 ± 0.06 | Timeout | **97.27 ± 0.30** |
| | **Avg. Rank** | 5.29 | 7.14 | 7.86 | 3.71 | 9.14 | 7.43 | 7.71 | 6.29 | 4.29 | 4.29 | **2.86** |
| Inductive | LastFM | 61.59 ± 5.72 | 60.62 ± 2.20 | 63.96 ± 0.41 | 65.48 ± 4.13 | 67.90 ± 0.44 | 77.37 ± 0.00 | 54.75 ± 1.31 | 59.98 ± 0.20 | 67.87 ± 0.53 | **78.70 ± 0.87** | 68.74 ± 0.55 |
| | Enron | 70.75 ± 0.69 | 67.37 ± 2.21 | 59.78 ± 1.12 | 73.22 ± 0.42 | 75.29 ± 0.66 | 75.00 ± 0.00 | 69.74 ± 1.19 | 70.72 ± 1.08 | 74.67 ± 0.80 | 75.40 ± 1.92 | **75.47 ± 1.41** |
| | MOOC | 67.53 ± 1.76 | 62.60 ± 1.27 | 74.44 ± 0.81 | 76.89 ± 2.13 | 70.08 ± 0.33 | 48.18 ± 0.00 | 71.80 ± 1.09 | 72.25 ± 0.57 | 80.78 ± 0.89 | 68.17 ± 3.73 | **81.08 ± 0.82** |
| | Reddit | 83.40 ± 0.33 | 82.75 ± 0.36 | 87.46 ± 0.10 | 84.57 ± 0.19 | 88.19 ± 0.20 | 85.93 ± 0.00 | 84.41 ± 0.18 | 82.24 ± 0.24 | 86.25 ± 0.64 | **91.42 ± 2.18** | 86.35 ± 0.52 |
| | Wikipedia | 70.41 ± 0.39 | 67.57 ± 0.94 | 81.54 ± 0.31 | 81.21 ± 0.30 | 68.48 ± 3.64 | 81.73 ± 0.00 | 73.51 ± 1.88 | 84.20 ± 0.36 | 64.09 ± 9.75 | **93.67 ± 0.11** | 75.64 ± 2.42 |
| | UCI | 64.14 ± 1.25 | 54.10 ± 2.74 | 59.60 ± 0.61 | 63.76 ± 0.99 | 57.85 ± 0.59 | 58.03 ± 0.00 | 65.46 ± 2.07 | **74.25 ± 0.71** | 64.92 ± 0.83 | 66.51 ± 0.25 | 66.83 ± 2.83 |
| | Social Evo. | 91.81 ± 1.69 | 92.77 ± 0.64 | 93.54 ± 0.48 | 94.86 ± 1.25 | 90.10 ± 0.11 | 87.88 ± 0.00 | 95.13 ± 0.83 | 94.50 ± 0.26 | 95.01 ± 0.15 | Timeout | **97.37 ± 0.26** |
| | **Avg. Rank** | 7.86 | 9.57 | 6.14 | 5.43 | 6.29 | 6.43 | 6.71 | 6.00 | 5.14 | 3.86 | **2.57** |

report the hyperparameter searching strategy as well as the best settings of DyGFormer and DyGMamba on synthetic datasets in Table 11. For all experiments with DyGMamba, we set the numbers of layers in both node-level and time-level SSMs as 2.

# D   Negative Edge Sampling Strategies during Evaluation

We justify why we do not do historical NSS for inductive link prediction. As described in Poursafaei et al. (2022), historical NSS focuses on sampling negative edges from the set of edges that have been observed during previous timestamps but are absent in the current step. In the setting of inductive link prediction, models are asked to predict the links between the nodes unseen in the training dataset. This means when doing historical NSS, models only need to care about the previously observed edges in the test set (or validation set during validation) for choosing negative edges. This makes historical NSS the same as inductive NSS in the inductive link prediction, where inductive NSS samples negative edges that have been observed only in the test set, but not training set. Empirical results shown in Appendix C.2 Table 13 and 14 of Yu et al. (2023) also prove that there is no difference between historical and inductive NSS in inductive link prediction. So we omit the results of historical NSS in our paper.

# E   AUC-ROC Results on Real-World Datasets

Table 12 and 13 presents the AUC-ROC results of all baselines and DyGMamba on real-world datasets. We have similar observations as the AP results shown in Table 1 and 2. DyGMamba still demonstrates superior performance and can achieve the best average rank under any NSS setting in both transductive and inductive link prediction.

Table 13: AUC-ROC of inductive dynamic link prediction. EdgeBank cannot do inductive link prediction so is not reported.

| NSS | Datasets | JODIE | DyRep | TGAT | TGN | CAWN | TCL | GraphMixer | DyGFormer | CTAN | DyGMamba |
|---|---|---|---|---|---|---|---|---|---|---|---|
| Random | LastFM | 82.49 ± 0.94 | 82.82 ± 1.17 | 76.76 ± 0.22 | 82.61 ± 2.62 | 87.92 ± 0.15 | 76.95 ± 1.34 | 80.34 ± 0.14 | 94.10 ± 0.09 | 61.49 ± 2.78 | **94.37 ± 0.13** |
| | Enron | 80.16 ± 1.50 | 75.82 ± 3.14 | 64.25 ± 1.29 | 79.40 ± 1.77 | 86.84 ± 0.89 | 81.03 ± 0.93 | 76.08 ± 0.92 | 89.59 ± 0.10 | 75.23 ± 2.24 | **89.76 ± 0.21** |
| | MOOC | 83.82 ± 0.30 | 83.42 ± 0.77 | 86.67 ± 0.24 | **91.58 ± 0.74** | 81.76 ± 0.46 | 82.42 ± 0.71 | 82.76 ± 0.13 | 87.75 ± 0.42 | 66.38 ± 1.59 | 89.34 ± 0.12 |
| | Reddit | 96.42 ± 0.13 | 95.87 ± 0.21 | 97.02 ± 0.04 | 97.30 ± 0.12 | 98.42 ± 0.01 | 94.63 ± 0.08 | 94.95 ± 0.08 | 98.70 ± 0.02 | 82.35 ± 4.03 | **98.88 ± 0.01** |
| | Wikipedia | 94.43 ± 0.28 | 91.31 ± 0.40 | 95.93 ± 0.19 | 97.71 ± 0.19 | 98.05 ± 0.03 | 97.03 ± 0.08 | 96.26 ± 0.04 | 98.49 ± 0.02 | 92.59 ± 0.70 | **98.72 ± 0.03** |
| | UCI | 78.78 ± 1.11 | 58.84 ± 2.54 | 77.41 ± 0.65 | 86.27 ± 1.49 | 90.27 ± 0.40 | 81.67 ± 1.01 | 89.26 ± 0.42 | 92.43 ± 0.20 | 48.58 ± 6.02 | **92.70 ± 0.19** |
| | Social Evo. | 93.62 ± 0.36 | 90.20 ± 2.05 | 93.52 ± 0.05 | 93.21 ± 0.90 | 84.73 ± 0.20 | 94.63 ± 0.06 | 94.09 ± 0.03 | 95.30 ± 0.05 | Timeout | **95.36 ± 0.04** |
| | **Avg. Rank** | 6.00 | 7.43 | 7.00 | 4.57 | 4.71 | 6.14 | 6.14 | 2.14 | 9.71 | **1.14** |
| Inductive | LastFM | 69.85 ± 1.70 | 68.14 ± 1.61 | 69.89 ± 0.41 | 67.01 ± 5.77 | 67.72 ± 0.20 | 63.15 ± 1.17 | 69.93 ± 0.17 | 69.86 ± 0.80 | 57.85 ± 3.67 | **70.59 ± 0.57** |
| | Enron | 65.95 ± 1.27 | 62.20 ± 2.15 | 56.52 ± 0.84 | 64.21 ± 0.94 | 62.07 ± 0.72 | 67.56 ± 1.34 | 67.39 ± 1.33 | 66.07 ± 0.65 | 68.70 ± 1.82 | **68.98 ± 1.00** |
| | MOOC | 65.37 ± 0.96 | 62.97 ± 2.05 | 74.94 ± 0.80 | 76.36 ± 2.91 | 71.18 ± 0.54 | 71.30 ± 1.21 | 72.15 ± 0.65 | 80.42 ± 0.72 | 58.06 ± 0.89 | **81.12 ± 0.63** |
| | Reddit | 61.84 ± 0.44 | 60.35 ± 0.53 | 64.92 ± 0.08 | 65.24 ± 0.08 | 65.37 ± 0.12 | 61.85 ± 0.11 | 64.56 ± 0.26 | 64.80 ± 0.53 | **81.70 ± 4.71** | 64.93 ± 0.89 |
| | Wikipedia | 61.66 ± 0.30 | 56.34 ± 0.67 | 78.40 ± 0.77 | 75.86 ± 0.50 | 59.00 ± 4.33 | 71.45 ± 2.23 | 82.76 ± 0.11 | 58.21 ± 8.78 | **91.12 ± 0.13** | 68.72 ± 2.23 |
| | UCI | 60.66 ± 1.82 | 51.50 ± 2.08 | 61.27 ± 0.78 | 62.07 ± 0.67 | 55.60 ± 1.22 | 65.87 ± 1.90 | **75.72 ± 0.70** | 64.37 ± 0.98 | 51.68 ± 2.60 | 66.95 ± 2.22 |
| | Social Evo. | 88.98 ± 0.81 | 86.43 ± 1.48 | 92.37 ± 0.50 | 91.66 ± 2.14 | 83.84 ± 0.21 | 95.50 ± 0.31 | 93.88 ± 0.22 | 94.97 ± 0.36 | Timeout | **96.65 ± 0.29** |
| | **Avg. Rank** | 7.00 | 8.71 | 5.14 | 5.14 | 7.14 | 5.14 | 3.57 | 4.71 | 6.14 | **2.29** |

# F Dynamic Node Classification

We first give the definition of the dynamic node classification task.

**Definition 3** (Dynamic Node Classification). *Given a CTDG $\mathcal{G}$, a source node $u \in \mathcal{N}$, a destination node $v \in \mathcal{N}$, a timestamp $t \in \mathcal{T}$, and all the interactions before $t$, i.e., $\{(u_i, v_i, t_i)|t_i < t, (u_i, v_i, t_i) \in \mathcal{G}\}$, dynamic node classification aims to predict the state (e.g., dynamic node label) of $u$ or $v$ at $t$ in the condition that the interaction $(u, v, t)$ exists.*

We follow Rossi et al. (2020); Xu et al. (2020); Yu et al. (2023) to conduct dynamic node classification by estimating the state of a node in a given interaction at a specific timestamp. A classification MLP is employed to map the node representations as well as the learned temporal patterns to the labels. AUC-ROC is used as the evaluation metric and we follow the dataset splits introduced in Yu et al. (2023) (70%15%/15% for training/validation/testing in chronological order) for node classification. Table 14 shows the node classification results on Wikipedia and Reddit (the only two CTDG datasets for dynamic node classfication), we observe that DyGMamba can achieve the best average rank, showing its strong performance. Note that both Wikipedia and Reddit are not long-range temporal dependent datasets, therefore we do not include this part into the main body of the paper. Nonetheless, DyGMamba's great results on these datasets further prove its strength in CTDG modeling, regardless of the type of the dataset (whether long-range temporal dependent or not).

Table 14: AUC-ROC of dynamic node classification.

| Datasets | JODIE | DyRep | TGAT | TGN | CAWN | TCL | GraphMixer | DyGFormer | CTAN | DyGMamba |
|---|---|---|---|---|---|---|---|---|---|---|
| Wikipedia | **88.10 ± 1.57** | 87.41 ± 1.94 | 83.42 ± 2.92 | 85.51 ± 3.28 | 84.59 ± 1.16 | 79.03 ± 1.18 | 85.60 ± 1.73 | 86.35 ± 2.19 | 87.38 ± 0.14 | 87.44 ± 0.82 |
| Reddit | 59.53 ± 3.18 | 63.12 ± 0.51 | **69.31 ± 2.18** | 63.21 ± 3.00 | 65.22 ± 0.79 | 68.04 ± 2.00 | 64.42 ± 1.15 | 67.67 ± 1.39 | 67.29 ± 0.15 | 67.70 ± 1.32 |
| **Avg. Rank** | 5.50 | 6.00 | 5.00 | 7.50 | 7.00 | 6.00 | 6.50 | 4.50 | 4.50 | **2.50** |

# G Efficieny Analysis Complete Results

We first provide the efficiency analysis results of all baselines in this section. We then provide a comparison of total training time among DyGFormer, CTAN and DyGMamba.

## G.1 Efficiency Statistics for all baselines

We provide the efficiency statistics for all baselines in Table 15. To supplement, in Figure 6, we further plot the comparison of model size, per epoch training time and GPU Memory across models on Reddit and Social Evo., analogous to the analysis in Sec. 4.3.1. CTAN cannot be trained on Social Evo. before timeout and

therefore does not appear in the plot of Social Evo.. The complete statistics including the numbers in the plots can be found in Table 15.

Table 15: Efficiency statistics for all baselines. EdgeBank is non-parameterized and not a machine learning model so we omit it here. # Params means number of parameters (MB). Time and Mem denote per epoch training time (min) and GPU memory (GB), respectively. The numbers in this table are the average results of five runs with different random seeds.

| Datasets | LastFM | | | Enron | | | MOOC | | | UCI | | | Reddit | | | Social Evo. | | |
|---|---|---|---|---|---|---|---|---|---|---|---|---|---|---|---|---|---|---|
| Models | # Params | Time | Mem | # Params | Time | Mem | # Params | Time | Mem | # Params | Time | Mem | # Params | Time | Mem | # Params | Time | Mem |
| JODIE | 0.75 | 4.4 | 2.28 | 0.75 | 0.07 | 1.30 | 0.75 | 0.78 | 2.36 | 0.75 | 0.03 | 1.44 | 0.75 | 3.95 | 1.10 | 0.75 | 4.70 | 1.71 |
| DyRep | 2.64 | 6.6 | 2.29 | 2.64 | 0.10 | 1.34 | 2.64 | 0.88 | 2.38 | 2.64 | 0.05 | 1.51 | 2.64 | 5.75 | 1.21 | 2.64 | 7.55 | 1.76 |
| TGAT | 4.02 | 22.75 | 4.15 | 4.02 | 1.28 | 3.46 | 4.02 | 4.08 | 3.64 | 4.02 | 0.60 | 3.42 | 4.02 | 16.33 | 2.98 | 4.02 | 25.50 | 3.89 |
| TGN | 3.68 | 12.14 | 2.21 | 3.68 | 0.15 | 1.45 | 3.68 | 1.03 | 2.54 | 3.68 | 0.08 | 1.51 | 3.67 | 2.05 | 1.67 | 3.67 | 3.83 | 1.78 |
| CAWN | 15.35 | 99.00 | 14.92 | 15.35 | 2.62 | 4.03 | 15.35 | 13.45 | 8.02 | 15.35 | 1.95 | 9.40 | 15.35 | 20.16 | 5.89 | 15.35 | 85.66 | 8.14 |
| TCL | 3.37 | 6.23 | 3.04 | 3.37 | 0.30 | 2.51 | 3.37 | 1.00 | 2.49 | 3.37 | 0.13 | 2.00 | 3.37 | 2.25 | 1.82 | 3.37 | 5.05 | 2.48 |
| GraphMixer | 2.45 | 16.35 | 2.78 | 2.45 | 1.20 | 2.23 | 2.45 | 4.02 | 2.40 | 2.45 | 0.73 | 2.19 | 2.44 | 4.92 | 1.57 | 2.45 | 15.50 | 2.71 |
| DyGFormer | 5.56 | 47.00 | 7.57 | 4.80 | 2.73 | 3.23 | 4.80 | 8.32 | 3.35 | 4.15 | 0.62 | 2.30 | 4.24 | 7.00 | 2.42 | 4.14 | 20.00 | 2.77 |
| CTAN | 0.45 | 3.33 | 1.44 | 0.47 | 0.50 | 1.33 | 0.68 | 3.22 | 2.30 | 0.50 | 0.38 | 1.30 | 0.53 | 0.86 | 1.54 | 0.45 | 2.41 | 0.63 |
| DyGMamba | 2.78 | 28.45 | 4.17 | 2.03 | 2.05 | 2.74 | 1.65 | 4.88 | 2.48 | 1.37 | 0.60 | 1.93 | 3.32 | 6.30 | 2.07 | 3.22 | 17.80 | 2.59 |

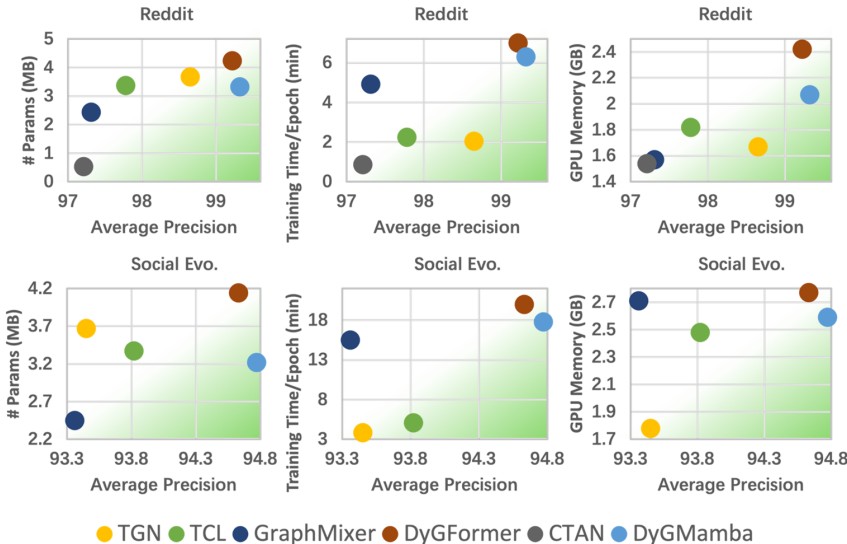

Figure 6: Efficiency comparison on Reddit and Social Evo. among DyGMamba and five baselines in terms of number (#) of parameters, training time per epoch and GPU memory.

## G.2 Total Training Time Comparison among DyGFormer, CTAN and DyGMamba

We present the per epoch training time, number of epochs until the best performance and the total training time in Table 16. Total training time computes the total amount of time a model requires to reach its maximum performance, without considering the patience during training. We observe that CTAN requires much more epochs to converge, e.g., on LastFM it uses almost 54 times of epochs than DyGMamba to reach its best performance.

## G.3 Modeling an Increasing Number of Temporal Neighbors with Limited Total Training Time

To further show DyGMamba's superior efficiency against baseline methods, we do the following experiments. We train five best performing models (as shown in Table 1) on Enron with a gradually increasing number of temporal neighbors[10] and report their performance. The number of sampled neighbors spans from 8, 16,

---

[10]For CTAN, by number of temporal neighbors we mean the sampler size in each graph convolutional layer, i.e., the size of the sampled temporal neighborhood for each node at a timestamp.

Table 16: Comparison among DyGFormer, CTAN and DyGMamba on per epoch training time ($T_{ep}$ (min)), number of epochs until the best performance (# Epoch) and the total training time ($T_{tot}$ (min)). $T_{tot} = T_{ep} \times$ # Epoch. The numbers in this table are the average results of five runs with different random seeds. CTAN cannot be trained on Social Evo. before timeout and thus without # Epoch and $T_{tot}$.

| Datasets | LastFM | | | Enron | | | MOOC | | | UCI | | | Reddit | | | Social Evo. | | |
|---|---|---|---|---|---|---|---|---|---|---|---|---|---|---|---|---|---|---|
| Models | $T_{ep}$ | # Epoch | $T_{tot}$ | $T_{ep}$ | # Epoch | $T_{tot}$ | $T_{ep}$ | # Epoch | $T_{tot}$ | $T_{ep}$ | # Epoch | $T_{tot}$ | $T_{ep}$ | # Epoch | $T_{tot}$ | $T_{ep}$ | # Epoch | $T_{tot}$ |
| DyGFormer | 47.00 | 49.60 | 2331.20 | 2.73 | 32.80 | 89.54 | 8.32 | 64.20 | 534.14 | 0.62 | 34.80 | 21.58 | 4.24 | 24.60 | 104.30 | 20.00 | 30.22 | 604.40 |
| CTAN | 3.33 | 635.00 | 2114.55 | 0.50 | 173.00 | 86.50 | 3.22 | 138.00 | 444.36 | 0.38 | 236.00 | 89.68 | 0.53 | 327.18 | 173.41 | 0.45 | Timeout | Timeout |
| DyGMamba | 28.45 | 11.80 | 335.71 | 2.05 | 33.00 | 67.65 | 4.88 | 38.00 | 185.44 | 0.60 | 28.00 | 16.80 | 3.32 | 26.80 | 88.98 | 17.80 | 24.40 | 434.32 |

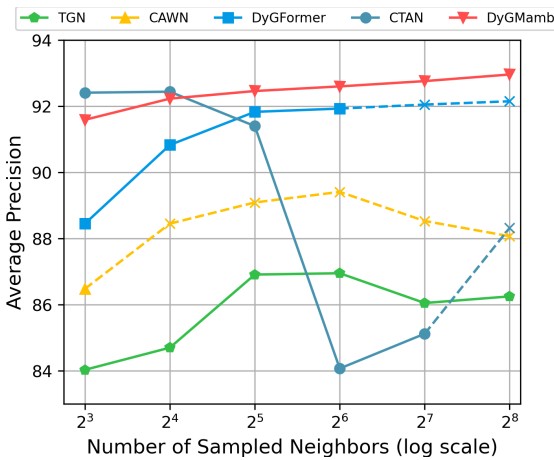

Figure 7: Performance comparison among TGN, CAWN, DyGFormer, CTAN and DyGMamba on Enron, with an increasing number of encoded temporal neighbors. The metric is AP under random NSS on transductive link prediction. The time limit for training is 120 min. If a model fails to complete training within this limit, a cross × is used to mark the data point. Dashed lines indicate that models start to exceed time limit as neighbor number increases.

32, 64, 128 to 256 (Note that these numbers are different from the best hyperparameters reported in Yu et al. (2023)). We fix the patch size $p$ of DyGFormer and DyGMamba to 1 in order to maximize their input sequence lengths. We set a time limit of 120 minutes for the total training time. We let all the experiments finish the complete training process and note down the ones that exceed the time limit. In this way, we not only care about the per epoch training time, but also pay attention to how long it takes for models to converge. The experimental results are reported in Fig. 7. The points marked with crosses (×) mean that the training process cannot finish within the time limit (although we still plot their corresponding performance). We find that only TGN and DyGMamba can successfully converge within the time limit when the number of considered neighbors increases to 256. DyGMamba can constantly achieve performance gain from modeling more temporal neighbors while TGN cannot. CAWN is extremely time consuming so it cannot finish training within the time limit even when it is asked to model 16 temporal neighbors. As for the methods designed for long-range temporal information propagation, DyGFormer and CTAN consume much longer total training time than DyGMamba. They fail to converge within 120 minutes when the number of considered neighbors reaches 128 and 256, respectively. We also observe that CTAN's performance fluctuates greatly with the increasing temporal neighbors, indicating that it is not stable to model a large number of temporal neighbors. This also implies that increasing the amount of historical information will gradually make CTAN harder to converge, which might cause trouble in modeling long range temporal dependent datasets.

## H  Details of S4 and Mamba Operations

**Single-Input Single-Output.**   Given a sequence of vector elements as input, SISO means that the SSM processes each input dimension in parallel with the same set of parameters. For example, a sequence of

$d_2$-dimensional vectors will be split into $d_2$ 1-dimensional sequences with the same sequence length. Each of them will be computed in parallel as in Eq. 2 with a shared set of SSM parameters. After computation, all these $d_2$ sequences will be rearranged back into a sequence of $d_2$-dimensional vectors. SISO fails to mix the information across dimensions of each vector. To address this, S4 and Mamba employs a mixing linear layer $f_{\text{mix}}(\cdot) : \mathbb{R}^{d_2} \to \mathbb{R}^{d_2}$ on each $d_2$-dimensional vector to mix the information across $d_2$ dimensions. For more details, please refer to (Gu et al., 2022b), (Smith et al., 2023) and (Gu & Dao, 2023).

**SSM Function.** $\text{SSM}_{\bar{\mathbf{A}}, \bar{\mathbf{B}}, \mathbf{C}}(\cdot)$ takes a matrix as input. The input matrix can be considered as a sequence of vector elements, where each row of the matrix corresponds to an element. The output of $\text{SSM}_{\bar{\mathbf{A}}, \bar{\mathbf{B}}, \mathbf{C}}(\cdot)$ is also a matrix, where each row of the output matrix is the output of its corresponding input vector element. $\text{SSM}_{\bar{\mathbf{A}}, \bar{\mathbf{B}}, \mathbf{C}}(\cdot)$ can be viewed as using S4 or Mamba to process a sequence of vectors in the SISO fashion, based on their parameters $\bar{\mathbf{A}}, \bar{\mathbf{B}}, \mathbf{C}$.

## I Motivation of Using SSM for Temporal Pattern Modeling

The biggest motivation of using SSM for temporal pattern modeling is that it helps to maintain good efficiency. If in the future we want to deploy DyGMamba on larger datasets that require much longer historical histories for modeling, the value of $k$ will also increase accordingly and the time difference sequence will not be short anymore. Besides, as we have chosen SSM to model historical one-hop temporal neighbors in the node-level SSM, it is natural to employ another SSM for temporal pattern modeling.

## J Impact of Patch Size on MOOC

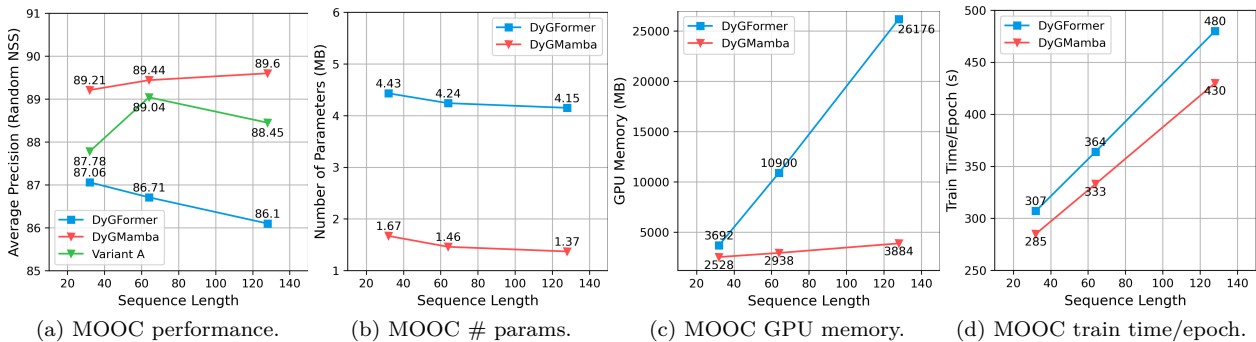

(a) MOOC performance.  (b) MOOC # params.  (c) MOOC GPU memory.  (d) MOOC train time/epoch.

Figure 8: Impact of patch size on DyGFormer, DyGMamba and Variant A, given a fixed number of sampled temporal neighbors $\rho$ on MOOC. Patch size $p$ varies from 4, 2, 1. Sequence length $\rho/p$ increases as patch size decreases. Performance is the transductive AP under random NSS.

Apart from the analysis on Enron, we further analyze the impact of patch size on another long-term temporal dependent dataset MOOC. Fig. 8b shows the numbers of parameters of DyGFormer and DyGmamba with different patch sizes on MOOC. We confirm that patching greatly affects model sizes. We decrease the patch size gradually from 4 to 1 (the optimal value of DyGMamba's hyperparameter $\rho$ & $p$ is 128 & 4) and track DyGMamba's performance (Fig. 8a) as well as efficiency (Fig. 8b to 8d) on MOOC. Meanwhile, we also keep track on DyGFormer under the same patch size for comparison. Same as our analysis on Enron, we plot the performance of Variant A under different patch sizes in Fig. 8a as well. **We can draw the same conclusions as in our analysis on Enron.**

## K Performance of DyGMamba on Discrete-Time Dynamic Graphs

To better benchmark DyGMamba, we test the performance of DyGMamba on 6 DTDG datasets collected in (Yu et al., 2023) (i.e., Flights, Can. Parl, US Legis., UN Trade, UN Vote and Contact) and compare it

with the 10 recent baselines discussed in Sec. 4.1 (i.e., JODIE, DyRep, TGAT, TGN, CAWN, EdgeBank, TCL, GraphMixer, DyGFormer, CTAN)[11]. The value of the hyperparameter $\rho$ & $p$ of DyGMamba is set as same as the one for DyGFormer. The optimal $\rho$ & $p$ for Flights, Can. Parl, US Legis., UN Trade, UN Vote and Contact is $\{256 \& 8, 2048 \& 64, 256 \& 8, 256 \& 8, 256 \& 8, 128 \& 4, 32 \& 1\}$. We report the best hyperparameter $k$ of DyGMamba for each dataset in Table 17. We use the implementations and the best hyperparameters provided by Yu et al. (2023) for all baseline models except CTAN. For CTAN, we use its official implementation, fixing the number of layers to 5. All models are trained with a batch size of 200 for fair efficiency analysis. We report the AP of both models on DTDG datasets in the transductive and inductive settings in Table 18 and Table 19, respectively. In addition, Table 20 and Table 21 present the AUC-ROC of both models in the transductive and inductive settings, respectively.

We find that DyGMamba achieves strong overall performance (Avg. Rank) on DTDGs in both transductive and inductive link prediction tasks. However, compared to its performance on CTDGs, DyGMamba's performance on DTDGs is not always highly competitive. This is expected, as DyGMamba is not originally designed for DTDGs and lacks the ability to optimally handle concurrent edges.

One interesting finding is that on the long-range temporal dependent dataset Can. Parl which requires sampling 2048 neighbors for modeling, DyGMamba can benefit from such long neighbor sequence and achieve the best performance among all models, indicating the importance of modeling long-term temporal information as well as the strong capability of our model in capturing it. More importantly, we find that DyGMamba can benefit from greater value of $k$ as the number of the sampled neighbors $\rho$ increases. For example, on Can. Parl, as shown in Table 17, the optimal value of $k$ is 100. This makes our selection of using Mamba for temporal pattern modeling more reasonable since Mamba can better demonstrate its advantage in efficiency when there is a growing time difference sequence corresponding to the temporal pattern.

Table 17: DyGMamba hyperparameter searching strategy on DTDG datasets. The best settings are marked as bold.

| Datasets | $k$ |
|---|---|
| Flights | $\{\mathbf{30}, 10, 5\}$ |
| Can. Parl. | $\{200, \mathbf{100}, 30\}$ |
| US Legis. | $\{\mathbf{30}, 10, 5\}$ |
| UN Trade | $\{\mathbf{30}, 10, 5\}$ |
| UN Vote | $\{30, \mathbf{10}, 5\}$ |
| Contact | $\{30, 10, \mathbf{5}\}$ |

# L    Detailed Discussion of Motivation

The goal of this work is to introduce a model capable of effective and efficient reasoning over CTDGs using extensive historical information. Our motivation is supported by the following considerations:

**Abundant Historical Information in Real-World Datasets.**    We quantify the availability of historical neighbors across various datasets for transductive and inductive link prediction tasks under random, historical, and inductive NSS settings, as shown in Table 22. Observations indicate that real-world CTDG datasets, such as LastFM and Enron, typically possess extensive one-hop historical information. Although it is feasible to perform predictions based solely on recent interactions, we posit that effectively leveraging longer historical contexts can further enhance model performance.

**Limitations of Existing Models in Handling Extensive Historical Information.**    Two recent state-of-the-art methods, DyGFormer and CTAN, explicitly designed to capture long-range historical dependencies, have been considered in our study. Our experimental results reveal distinct limitations: DyGFormer

---

[11]Although we have discussed in Sec. 7 that DyGMamba is not suitable to reason over DTDGs, we still benchmark our model on them to show its effectiveness.

Table 18: AP of transductive dynamic link prediction on DTDGs. The best and the second best results are marked as **bold** and underlined, respectively.

| NSS | Datasets | JODIE | DyRep | TGAT | TGN | CAWN | EdgeBank | TCL | GraphMixer | DyGFormer | CTAN | DyGMamba |
|---|---|---|---|---|---|---|---|---|---|---|---|---|
| Random | Can. Parl. | 69.58 ± 0.28 | 68.94 ± 3.55 | 71.57 ± 0.68 | 69.37 ± 1.86 | 69.07 ± 3.01 | 64.55 ± 0.00 | 71.49 ± 0.46 | 77.49 ± 0.62 | 97.63 ± 0.21 | 87.01 ± 1.10 | **99.57 ± 0.08** |
| | Contact | 94.75 ± 0.82 | 95.97 ± 0.16 | 96.55 ± 0.11 | 97.21 ± 0.44 | 90.65 ± 0.16 | 92.58 ± 0.00 | 94.67 ± 0.11 | 91.88 ± 0.07 | 98.30 ± 0.00 | **98.53 ± 0.09** | 98.43 ± 0.12 |
| | Flights | 96.03 ± 1.17 | 95.49 ± 0.49 | 93.92 ± 0.04 | 98.01 ± 0.39 | 98.50 ± 0.01 | 89.35 ± 0.00 | 91.30 ± 0.05 | 91.02 ± 0.04 | 98.92 ± 0.01 | 93.64 ± 0.54 | **98.95 ± 0.05** |
| | UN Trade | 64.43 ± 0.25 | 62.58 ± 0.93 | 61.53 ± 0.18 | 64.03 ± 0.69 | 65.41 ± 0.04 | 60.41 ± 0.00 | 62.19 ± 0.07 | 62.87 ± 0.12 | 66.69 ± 0.33 | **69.01 ± 0.38** | 67.50 ± 0.24 |
| | UN Vote | 63.35 ± 0.99 | 63.90 ± 0.08 | 52.89 ± 0.28 | 65.18 ± 0.61 | 52.77 ± 0.07 | 58.49 ± 0.00 | 51.89 ± 0.18 | 52.18 ± 0.13 | 55.90 ± 0.24 | **70.19 ± 1.26** | 56.39 ± 0.18 |
| | US Legis. | 74.29 ± 0.68 | 73.81 ± 2.29 | 67.85 ± 2.80 | 76.46 ± 0.37 | 70.60 ± 0.25 | 58.39 ± 0.00 | 70.99 ± 0.25 | 72.44 ± 0.29 | 71.06 ± 1.06 | **79.44 ± 2.28** | 71.75 ± 0.26 |
| | Avg. Rank | 5.17 | 6.17 | 7.50 | 4.33 | 7.50 | 9.67 | 8.50 | 7.67 | 4.00 | 2.50 | 3.00 |
| Historical | Can. Parl. | 51.66 ± 0.51 | 62.46 ± 2.73 | 68.87 ± 0.96 | 66.49 ± 0.47 | 65.58 ± 3.34 | 63.76 ± 0.00 | 68.92 ± 2.04 | 74.58 ± 1.42 | 95.72 ± 1.84 | 89.28 ± 1.24 | **99.77 ± 0.12** |
| | Contact | 94.33 ± 1.77 | 96.27 ± 0.17 | 96.75 ± 0.72 | 96.39 ± 0.47 | 84.61 ± 0.52 | 88.83 ± 0.00 | 95.26 ± 0.19 | 93.29 ± 0.47 | 97.59 ± 0.18 | **97.86 ± 0.15** | 97.61 ± 0.04 |
| | Flights | 66.56 ± 1.39 | 67.53 ± 1.79 | 72.60 ± 0.33 | 67.77 ± 1.08 | 64.60 ± 0.64 | 70.53 ± 0.00 | 71.13 ± 0.19 | 71.40 ± 0.32 | 65.63 ± 0.21 | **75.75 ± 8.04** | 67.80 ± 2.17 |
| | UN Trade | 61.38 ± 1.27 | 58.56 ± 1.04 | 54.64 ± 0.22 | 56.64 ± 3.43 | 57.09 ± 1.80 | **81.08 ± 0.00** | 56.80 ± 1.20 | 57.91 ± 0.81 | 62.86 ± 1.66 | 67.38 ± 1.08 | 65.10 ± 0.02 |
| | UN Vote | 71.11 ± 1.00 | 70.33 ± 1.21 | 53.67 ± 2.31 | 69.16 ± 1.82 | 51.29 ± 0.41 | **84.76 ± 0.00** | 53.78 ± 2.16 | 54.17 ± 1.08 | 60.59 ± 0.94 | 74.75 ± 1.31 | 61.07 ± 0.39 |
| | US Legis. | 47.20 ± 1.72 | 81.60 ± 7.14 | 76.28 ± 4.16 | 67.45 ± 6.79 | 73.25 ± 14.99 | 63.31 ± 0.00 | 83.27 ± 0.88 | **86.05 ± 1.77** | 63.99 ± 11.58 | 84.20 ± 0.95 | 82.15 ± 1.02 |
| | Avg. Rank | 7.83 | 6.50 | 6.50 | 7.00 | 9.33 | 6.00 | 6.17 | 5.33 | 5.83 | 1.83 | 3.67 |
| Inductive | Can. Parl. | 48.13 ± 0.42 | 57.90 ± 2.61 | 69.54 ± 0.66 | 63.58 ± 1.59 | 66.94 ± 1.60 | 62.20 ± 0.00 | 68.23 ± 1.42 | 70.00 ± 0.57 | 95.01 ± 0.80 | 87.42 ± 0.71 | **98.32 ± 0.34** |
| | Contact | 92.34 ± 1.91 | 93.99 ± 0.24 | 95.14 ± 1.19 | 94.36 ± 0.88 | 89.61 ± 0.36 | 85.19 ± 0.00 | 92.08 ± 0.32 | 90.64 ± 0.56 | 94.93 ± 0.54 | **97.31 ± 0.19** | 95.43 ± 0.17 |
| | Flights | 69.39 ± 2.19 | 71.32 ± 2.44 | 75.68 ± 0.37 | 72.27 ± 1.33 | 69.01 ± 0.35 | **81.07 ± 0.00** | 75.10 ± 0.08 | 74.68 ± 0.35 | 70.31 ± 1.00 | 76.20 ± 7.96 | 73.79 ± 5.69 |
| | UN Trade | 60.55 ± 1.20 | 60.15 ± 0.52 | 58.86 ± 0.38 | 59.52 ± 3.79 | 63.37 ± 2.14 | **73.00 ± 0.00** | 61.86 ± 1.42 | 62.03 ± 0.63 | 53.53 ± 0.27 | 68.98 ± 0.73 | 58.89 ± 0.98 |
| | UN Vote | 67.81 ± 0.75 | 68.69 ± 0.24 | 52.66 ± 1.18 | 67.58 ± 1.51 | 51.90 ± 1.40 | 66.31 ± 0.00 | 49.17 ± 4.04 | 51.28 ± 1.25 | 53.06 ± 0.23 | **72.85 ± 1.36** | 52.24 ± 0.95 |
| | US Legis. | 46.58 ± 1.19 | 79.16 ± 7.11 | 76.09 ± 4.00 | 61.85 ± 7.02 | 71.78 ± 14.33 | 64.83 ± 0.00 | 74.33 ± 2.79 | **83.71 ± 0.69** | 62.01 ± 10.94 | 79.13 ± 2.03 | 81.67 ± 2.16 |
| | Avg. Rank | 8.00 | 6.00 | 5.50 | 7.00 | 7.83 | 5.83 | 6.67 | 5.50 | 6.83 | 2.17 | 4.67 |

Table 19: AP of inductive dynamic link prediction on DTDGs. EdgeBank cannot do inductive link prediction so is not reported.

| NSS | Datasets | JODIE | DyRep | TGAT | TGN | CAWN | TCL | GraphMixer | DyGFormer | CTAN | DyGMamba |
|---|---|---|---|---|---|---|---|---|---|---|---|
| Random | Can. Parl. | 53.69 ± 1.27 | 54.28 ± 1.84 | 55.60 ± 0.57 | 53.47 ± 1.04 | 55.33 ± 0.98 | 55.06 ± 0.67 | 56.69 ± 0.04 | 87.14 ± 1.40 | 62.97 ± 0.34 | **93.46 ± 2.62** |
| | Contact | 95.20 ± 0.34 | 92.32 ± 0.20 | 96.18 ± 0.21 | 92.74 ± 2.86 | 89.98 ± 0.36 | 93.93 ± 0.14 | 90.51 ± 0.03 | 98.04 ± 0.02 | 76.19 ± 5.71 | **98.16 ± 0.03** |
| | Flights | 94.45 ± 0.89 | 92.85 ± 0.55 | 88.70 ± 0.02 | 95.03 ± 1.04 | 97.07 ± 0.02 | 83.54 ± 0.10 | 83.03 ± 0.10 | 97.79 ± 0.04 | 75.12 ± 5.52 | **97.85 ± 0.22** |
| | UN Trade | 58.94 ± 0.28 | 56.46 ± 0.60 | 61.20 ± 0.21 | 56.97 ± 1.53 | 65.30 ± 0.14 | 62.24 ± 0.27 | 62.55 ± 0.11 | 64.62 ± 0.58 | 53.83 ± 0.20 | **70.55 ± 0.04** |
| | UN Vote | 55.65 ± 1.32 | 55.53 ± 1.43 | 52.38 ± 0.92 | **57.86 ± 1.72** | 49.66 ± 0.77 | 53.38 ± 0.60 | 51.94 ± 1.32 | 56.40 ± 0.16 | 55.02 ± 3.04 | 56.61 ± 0.13 |
| | US Legis. | 54.36 ± 1.90 | 57.54 ± 1.43 | 50.21 ± 1.77 | **59.39 ± 2.30** | 52.79 ± 0.41 | 57.12 ± 0.22 | 54.68 ± 2.07 | 54.39 ± 1.63 | 52.55 ± 3.42 | 55.95 ± 1.16 |
| | Avg. Rank | 6.00 | 6.17 | 6.50 | 6.00 | 6.33 | 5.83 | 6.50 | 3.00 | 8.00 | 1.67 |
| Inductive | Can. Parl. | 51.91 ± 1.35 | 52.52 ± 1.85 | 57.42 ± 0.57 | 53.55 ± 1.36 | 57.56 ± 0.81 | 56.59 ± 1.22 | 56.91 ± 0.55 | 84.72 ± 2.58 | 63.27 ± 1.07 | **92.68 ± 0.97** |
| | Contact | 91.22 ± 0.18 | 89.18 ± 0.48 | **94.93 ± 0.97** | 88.74 ± 2.60 | 74.88 ± 0.81 | 91.41 ± 0.40 | 89.67 ± 0.66 | 93.78 ± 0.88 | 70.10 ± 5.03 | 94.05 ± 0.32 |
| | Flights | 60.79 ± 1.25 | 62.54 ± 1.73 | 65.23 ± 0.42 | 59.58 ± 1.56 | 56.76 ± 0.11 | 65.12 ± 0.09 | 65.26 ± 0.44 | 56.45 ± 0.17 | **66.53 ± 2.41** | 57.76 ± 2.06 |
| | UN Trade | 55.33 ± 0.88 | 54.52 ± 0.52 | 54.21 ± 0.24 | 51.84 ± 1.57 | **56.29 ± 1.86** | 55.82 ± 1.21 | 55.73 ± 0.20 | 51.92 ± 0.13 | 51.91 ± 0.57 | 52.81 ± 0.18 |
| | UN Vote | 59.24 ± 1.60 | 61.97 ± 1.82 | 52.52 ± 2.93 | **66.41 ± 1.80** | 47.35 ± 0.75 | 56.34 ± 2.25 | 48.15 ± 0.58 | 52.79 ± 0.84 | 55.49 ± 0.78 | 53.70 ± 2.40 |
| | US Legis. | 55.59 ± 2.38 | 59.60 ± 1.64 | 50.85 ± 2.59 | **60.19 ± 2.60** | 55.15 ± 1.55 | 58.56 ± 0.99 | 55.15 ± 0.27 | 54.76 ± 2.35 | 52.65 ± 1.99 | 57.85 ± 0.23 |
| | Avg. Rank | 5.50 | 5.00 | 5.50 | 5.83 | 6.50 | 4.00 | 5.33 | 6.33 | 6.17 | 4.67 |

Table 20: AUC-ROC of transductive dynamic link prediction on DTDGs.

| NSS | Datasets | JODIE | DyRep | TGAT | TGN | CAWN | EdgeBank | TCL | GraphMixer | DyGFormer | CTAN | DyGMamba |
|---|---|---|---|---|---|---|---|---|---|---|---|---|
| Random | Can. Parl. | 78.34 ± 0.24 | 76.69 ± 4.51 | 76.36 ± 0.95 | 75.66 ± 1.62 | 74.24 ± 3.64 | 64.14 ± 0.00 | 76.64 ± 0.39 | 83.22 ± 0.84 | 98.00 ± 0.22 | 86.77 ± 1.17 | **99.69 ± 0.06** |
| | Contact | 96.36 ± 0.48 | 96.46 ± 0.14 | 97.18 ± 0.09 | 97.76 ± 0.35 | 90.45 ± 0.24 | 94.34 ± 0.00 | 95.53 ± 0.06 | 93.92 ± 0.01 | 98.52 ± 0.00 | **98.88 ± 0.06** | 98.68 ± 0.02 |
| | Flights | 96.48 ± 1.17 | 96.10 ± 0.50 | 94.60 ± 0.03 | 98.26 ± 0.36 | 98.45 ± 0.01 | 90.23 ± 0.00 | 91.26 ± 0.03 | 91.14 ± 0.03 | 98.93 ± 0.01 | 94.57 ± 0.57 | **98.98 ± 0.05** |
| | UN Trade | 69.10 ± 0.45 | 66.92 ± 0.53 | 63.99 ± 0.18 | 67.80 ± 0.81 | 68.57 ± 0.14 | 66.75 ± 0.00 | 64.72 ± 0.01 | 65.97 ± 0.10 | 70.53 ± 0.34 | **72.00 ± 0.24** | 71.41 ± 0.21 |
| | UN Vote | 68.26 ± 1.19 | 68.72 ± 0.08 | 53.25 ± 0.28 | 68.92 ± 0.72 | 53.15 ± 0.03 | 62.97 ± 0.00 | 51.73 ± 0.25 | 52.60 ± 0.09 | 57.65 ± 0.48 | **73.91 ± 1.08** | 58.48 ± 0.12 |
| | US Legis. | 82.13 ± 0.30 | 80.98 ± 1.92 | 75.00 ± 2.19 | 83.65 ± 0.27 | 77.10 ± 0.14 | 62.57 ± 0.00 | 77.11 ± 0.59 | 79.19 ± 0.23 | 77.65 ± 0.95 | **86.01 ± 1.68** | 79.03 ± 0.26 |
| | Avg. Rank | 4.83 | 5.33 | 8.00 | 5.67 | 8.83 | 9.83 | 9.00 | 7.83 | 3.00 | 2.67 | 2.00 |
| Historical | Can. Parl. | 62.08 ± 0.96 | 70.62 ± 3.57 | 72.11 ± 0.93 | 71.43 ± 0.99 | 70.44 ± 3.69 | 62.94 ± 0.00 | 74.07 ± 1.60 | 78.55 ± 0.72 | 96.46 ± 1.93 | 89.06 ± 1.14 | **99.82 ± 0.10** |
| | Contact | 96.23 ± 0.67 | 95.89 ± 0.14 | 96.01 ± 0.61 | 96.34 ± 0.24 | 83.63 ± 0.49 | 85.86 ± 0.00 | 94.70 ± 0.18 | 92.98 ± 0.39 | 97.20 ± 0.13 | **98.37 ± 0.10** | 97.27 ± 0.06 |
| | Flights | 68.84 ± 0.70 | 69.23 ± 1.44 | 72.33 ± 0.23 | 69.12 ± 0.81 | 65.93 ± 0.46 | 74.64 ± 0.00 | 70.81 ± 0.07 | 70.78 ± 0.32 | 67.50 ± 0.66 | **78.87 ± 7.94** | 68.98 ± 0.81 |
| | UN Trade | 68.89 ± 1.20 | 63.54 ± 1.02 | 59.33 ± 0.58 | 61.61 ± 2.98 | 64.74 ± 2.21 | **86.44 ± 0.00** | 62.11 ± 1.03 | 64.50 ± 0.17 | 72.17 ± 1.68 | 69.75 ± 0.90 | 71.41 ± 0.21 |
| | UN Vote | 77.54 ± 1.09 | 76.32 ± 1.04 | 55.61 ± 2.41 | 73.31 ± 1.65 | 50.95 ± 0.72 | **89.53 ± 0.00** | 54.70 ± 1.78 | 56.29 ± 1.46 | 64.79 ± 1.13 | 78.62 ± 1.36 | 65.17 ± 1.24 |
| | US Legis. | 54.26 ± 3.67 | 88.47 ± 4.92 | 83.49 ± 4.72 | 79.29 ± 4.52 | 80.85 ± 9.96 | 67.50 ± 0.00 | 86.20 ± 0.84 | **90.38 ± 0.73** | 75.60 ± 9.12 | 89.88 ± 0.54 | 88.36 ± 1.78 |
| | Avg. Rank | 7.33 | 6.00 | 6.83 | 6.83 | 9.17 | 5.67 | 6.83 | 5.67 | 5.50 | 2.17 | 4.00 |
| Inductive | Can. Parl. | 52.09 ± 0.28 | 63.40 ± 3.74 | 72.42 ± 0.82 | 67.47 ± 1.82 | 71.66 ± 1.91 | 61.25 ± 0.00 | 72.29 ± 1.43 | 71.25 ± 0.46 | 95.78 ± 0.85 | 85.99 ± 0.95 | **99.56 ± 0.21** |
| | Contact | 94.24 ± 0.24 | 94.11 ± 0.17 | 94.76 ± 0.98 | 94.70 ± 0.39 | 88.26 ± 0.29 | 85.86 ± 0.00 | 92.28 ± 0.24 | 90.71 ± 0.43 | 95.12 ± 0.28 | **97.93 ± 0.12** | 95.68 ± 0.20 |
| | Flights | 70.09 ± 1.39 | 71.38 ± 1.84 | 73.56 ± 0.22 | 72.41 ± 0.71 | 69.61 ± 0.26 | **81.10 ± 0.00** | 72.77 ± 0.04 | 72.03 ± 0.35 | 69.33 ± 0.56 | 79.82 ± 7.80 | 71.16 ± 3.24 |
| | UN Trade | 66.91 ± 1.17 | 65.40 ± 0.56 | 64.50 ± 0.93 | 64.50 ± 3.18 | 72.63 ± 2.17 | **74.25 ± 0.00** | 72.77 ± 0.00 | 68.28 ± 0.85 | 59.98 ± 0.40 | 69.74 ± 1.54 | 67.60 ± 0.64 |
| | UN Vote | 73.82 ± 1.05 | 74.67 ± 0.22 | 52.88 ± 1.86 | 72.60 ± 1.62 | 52.05 ± 1.65 | 72.87 ± 0.00 | 50.25 ± 7.46 | 51.37 ± 1.05 | 55.36 ± 0.30 | **76.64 ± 1.02** | 54.09 ± 0.06 |
| | US Legis. | 53.13 ± 2.57 | 86.74 ± 5.01 | 83.16 ± 3.65 | 73.50 ± 5.46 | 79.80 ± 10.61 | 68.72 ± 0.00 | 80.53 ± 3.50 | **88.82 ± 0.32** | 73.31 ± 9.22 | 86.09 ± 1.11 | 86.06 ± 2.27 |
| | Avg. Rank | 7.83 | 5.83 | 5.50 | 6.67 | 7.33 | 6.17 | 6.50 | 6.17 | 7.00 | 2.17 | 4.67 |

Table 21: AUC-ROC of inductive dynamic link prediction on DTDGs. EdgeBank cannot do inductive link prediction so is not reported.

| NSS | Datasets | JODIE | DyRep | TGAT | TGN | CAWN | TCL | GraphMixer | DyGFormer | CTAN | DyGMamba |
|---|---|---|---|---|---|---|---|---|---|---|---|
| Random | Can. Parl. | 53.62 ± 1.28 | 55.89 ± 1.77 | 56.67 ± 0.60 | 55.38 ± 2.11 | 57.91 ± 0.84 | 56.92 ± 0.97 | 59.29 ± 0.04 | 89.06 ± 0.39 | 59.13 ± 0.36 | **94.02 ± 3.02** |
| | Contact | 95.96 ± 0.20 | 92.00 ± 0.16 | 96.78 ± 0.17 | 94.04 ± 2.09 | 89.60 ± 0.32 | 94.82 ± 0.09 | 92.81 ± 0.02 | 98.30 ± 0.00 | 80.41 ± 4.30 | **98.44 ± 0.05** |
| | Flights | 94.82 ± 0.85 | 93.64 ± 0.44 | 88.61 ± 0.12 | 95.81 ± 0.87 | 96.87 ± 0.02 | 82.56 ± 0.13 | 82.28 ± 0.08 | 97.80 ± 0.05 | 79.41 ± 3.79 | **97.98 ± 0.25** |
| | UN Trade | 61.34 ± 0.38 | 58.41 ± 0.54 | 62.68 ± 0.21 | 58.71 ± 1.69 | 67.21 ± 0.15 | 63.83 ± 0.16 | 63.90 ± 0.02 | 67.28 ± 0.54 | 54.62 ± 0.93 | **68.26 ± 0.26** |
| | UN Vote | 56.24 ± 1.56 | 56.72 ± 1.90 | 52.44 ± 1.09 | **59.42 ± 2.12** | 47.92 ± 1.19 | 51.94 ± 0.69 | 51.65 ± 1.10 | 57.47 ± 0.27 | 53.40 ± 3.51 | 56.91 ± 0.12 |
| | US Legis. | 57.23 ± 1.92 | 60.86 ± 1.59 | 46.86 ± 3.44 | **62.36 ± 2.04** | 50.81 ± 1.02 | 58.02 ± 1.37 | 54.98 ± 2.39 | 52.96 ± 1.71 | 53.89 ± 3.55 | 57.17 ± 0.20 |
| | Avg. Rank | 5.83 | 6.17 | 6.67 | 4.83 | 6.50 | 5.83 | 6.33 | 3.00 | 7.83 | 2.00 |
| Inductive | Can. Parl. | 50.56 ± 2.02 | 52.07 ± 1.53 | 58.91 ± 0.48 | 54.61 ± 2.40 | 60.17 ± 0.46 | 58.37 ± 1.15 | 58.59 ± 0.77 | 86.13 ± 2.37 | 59.68 ± 1.05 | **92.37 ± 0.18** |
| | Contact | 91.20 ± 0.17 | 88.90 ± 0.26 | **94.44 ± 0.86** | 89.78 ± 2.06 | 75.38 ± 0.50 | 91.78 ± 0.28 | 89.82 ± 0.51 | 94.31 ± 0.43 | 74.41 ± 3.94 | 94.35 ± 0.29 |
| | Flights | 60.58 ± 0.50 | 61.64 ± 1.60 | 63.61 ± 0.32 | 59.28 ± 0.95 | 56.51 ± 0.03 | 63.64 ± 0.11 | 63.07 ± 0.35 | 55.67 ± 0.48 | **71.35 ± 1.86** | 56.58 ± 2.12 |
| | UN Trade | 58.51 ± 1.05 | 56.96 ± 0.44 | 58.71 ± 0.43 | 54.45 ± 1.33 | **62.65 ± 2.62** | 61.14 ± 1.09 | 61.00 ± 0.35 | 55.80 ± 0.11 | 52.05 ± 0.59 | 57.58 ± 0.20 |
| | UN Vote | 62.88 ± 1.52 | 65.80 ± 2.47 | 51.79 ± 3.30 | **71.29 ± 2.01** | 46.61 ± 1.35 | 57.35 ± 2.24 | 45.75 ± 1.04 | 54.62 ± 0.50 | 54.31 ± 1.14 | 54.83 ± 2.17 |
| | US Legis. | 59.83 ± 2.06 | **65.54 ± 1.42** | 47.15 ± 4.26 | 63.95 ± 2.30 | 53.14 ± 2.15 | 59.73 ± 1.06 | 56.22 ± 2.36 | 53.57 ± 2.73 | 53.64 ± 2.12 | 57.91 ± 3.41 |
| | Avg. Rank | 5.33 | 5.33 | 5.17 | 5.67 | 6.67 | 3.83 | 5.83 | 6.17 | 6.50 | 4.50 |

suffers from high computational complexity, while CTAN exhibits comparatively weaker predictive performance. Consequently, neither model achieves an optimal balance between effectiveness and computational efficiency. Our proposed model, DyGMamba, overcomes these limitations by demonstrating superior predictive capability and efficiency. Extensive experiments position DyGMamba as the top ranked method across multiple benchmark datasets.

**Proven Performance Enhancement with Increased Historical Information.** In Appendix G.3, we have included an experiment showing different models' performance with a varying number of temporal neighbors. From Figure 7, we observe that, for datasets with long-range temporal dependencies (such as Enron), an increase in sampled neighbors positively impacts model performance, particularly for DyGMamba and DyGFormer. We also find that for other baseline models, increasing the number of sampled neighbors does not consistently lead to performance improvements. This suggests that the key issue is not whether increasing sampled neighbors is always beneficial for dynamic graph modeling, but rather whether a model is robust enough to effectively process larger amounts of historical information. To further support this, we conducted additional experiments on Can. Parl[12] using DyGMamba with an increasing number of sampled temporal neighbors (64, 128, 256, 512, 1024 and 2048). The results, presented in Table 23, confirm that DyGMamba consistently benefits from an increasing neighbor count until reaching the large number of 2048, further demonstrating its ability to leverage long-range temporal dependencies.

**Temporal Pattern Modeling is Game Changer.** As mentioned in introduction, we wish to capture edge-specific temporal patterns for better CTDG modeling. The motivation can be well supported with the following experiment. We provide a comparison between DyGMamba, DyGFormer, and a modified version of DyGFormer where Transformer is replaced with Mamba (referred to as Variant G). Our results in Table 24 show a performance drop for Variant G across all datasets compared to DyGFormer. Additionally, Variant G performs significantly worse than DyGMamba, with particularly notable declines on the synthetic datasets S1, S2, and S3. We attribute this to two reasons:

- Intrinsic Limitations of Mamba: Previous study (Waleffe et al., 2024) has shown that Mamba is generally less effective than Transformer in tasks requiring strong copying or in-context learning abilities. In CTDG modeling, strong copying ability is crucial, as many predictions are based on re-calling repeated edges. Consequently, replacing Transformer with Mamba in DyGFormer inevitably leads to a performance drop, despite the gains in efficiency.

- The Importance of Temporal Pattern Modeling: Our dynamic information selection module plays a crucial role in enabling a Mamba-based model to outperform a Transformer-based model in CTDG modeling, especially when clear temporal patterns exist. This suggests that effectively capturing temporal patterns can significantly enhance the competitiveness of Mamba-based models in this context, further validating the novelty and the contribution of our work.

---

[12]Although Can. Parl is a DTDG, it requires huge number of sampled historical neighbors for optimal modeling. Therefore, it is considered here to prove the performance enhancement with increased historical information.

To summarize, the design our DyGMamba achieves a good balance between efficiency and effectiveness, thanks to temporal pattern modeling.

Table 22: Number of available historical neighbors for all datasets on both transductive and inductive link prediction under random/historical/inductive NSS settings. Trans LP and Ind LP denote transductive and inductive link prediction, respectively.

| NSS | Random NSS | | | | Historical NSS | | Inductive NSS | | | |
|---|---|---|---|---|---|---|---|---|---|---|
| | Trans LP | | Ind LP | | Trans LP | | Trans LP | | Ind LP | |
| Datasets | Avg. | Max | Avg. | Max | Avg. | Max | Avg. | Max | Avg. | Max |
| LastFM | 2,253.03 | 51,767 | 2,333.75 | 51,767 | 2,393.71 | 51,767 | 2,237.29 | 51,767 | 2,309.49 | 51,767 |
| Enron | 1,681.80 | 21,512 | 1,693.40 | 21,511 | 1,734.19 | 21,512 | 1,675.21 | 21,512 | 1,670.65 | 21,511 |
| MOOC | 2,304.04 | 19,473 | 2,312.62 | 19,473 | 3,265.62 | 19,473 | 2,293.64 | 19,473 | 2,275.23 | 19,473 |
| Reddit | 4,129.77 | 58,726 | 4,196.20 | 58,726 | 4,604.99 | 58,726 | 4,128.88 | 58,726 | 4,186.38 | 58,726 |
| Wikipedia | 144.39 | 1,937 | 149.67 | 1,937 | 164.49 | 1,937 | 144.17 | 1,937 | 149.22 | 1,937 |
| UCI | 172.82 | 1,546 | 193.87 | 1,546 | 261.31 | 1,546 | 173.08 | 1,546 | 192.72 | 1,546 |
| Social Evo. | 63,962.35 | 124,565 | 68,701.32 | 124,565 | 60,403.62 | 124,565 | 60,395.83 | 124,565 | 65,191.24 | 124,565 |

Table 23: DyGMamba's performance with increasing sampled neighbors on transductive link prediction under random NSS.

| Can. Parl | 64 | 128 | 256 | 512 | 1024 | 2048 |
|---|---|---|---|---|---|---|
| DyGMamba | 72.58 | 74.46 | 76.85 | 82.67 | 94.88 | 99.57 |

Table 24: Comparison among Variant G, DyGFormer and DyGMamba on transductive link prediction under random NSS.

| Models | LastFM | Enron | MOOC | Reddit | Wikipedia | UCI | Social Evo. | S1 | S2 | S3 |
|---|---|---|---|---|---|---|---|---|---|---|
| Variant G | 92.69 | 92.24 | 87.13 | 98.89 | 98.70 | 95.32 | 94.29 | 54.40 | 55.13 | 72.53 |
| DyGFormer | 92.95 | 92.42 | 87.66 | 99.22 | 99.03 | 95.74 | 94.63 | 55.19 | 57.80 | 79.20 |
| DyGMmaba | **93.35** | **92.65** | **89.21** | **99.32** | **99.15** | **95.91** | **94.77** | **81.85** | **85.36** | **86.59** |

