# OpenReview forum: "DyGMamba: Efficiently Modeling Long-Term Temporal Dependency on Continuous-Time Dynamic Graphs with State Space Models"
_TMLR — Accepted by TMLR_

### Review · Reviewer_TVZU · 2025-02-19

**Summary Of Contributions:**

This paper studies representation learning in continuous-time dynamic graphs and introduces a new method, DyGMamba. Leveraging Mamba as the backbone, this method effectively encodes long historical neighbor information while maintaining efficiency. Specifically, it stacks node-level and time-level blocks to capture both types of information, which are then combined for downstream predictions. Extensive experiments demonstrate promising results.

**Audience:**

Yes

**Claims And Evidence:**

No

**Requested Changes:**

I recommend that the authors further refine their discussion of the motivation.

**Strengths And Weaknesses:**

***S1*** The overall writing is clear and easy to follow, particularly in the introduction.

***S2*** The experiments are extensive, including complexity analysis and efficiency studies.

---

***W1*** Limited novelty and effectiveness. The methodology of DyGMamba appears to be a trivial combination of DyGFormer’s design (e.g., the feature encoder, patching) with the existing Mamba architecture. I wonder whether replacing the backbone Transformer in DyGFormer with Mamba would yield comparable performance to DyGMamba. If so, adopting Mamba as the new backbone for continuous-time dynamic graphs (CTDG) would be more of a technical adjustment than a significant research contribution. Despite improvements in efficiency, Tables 1 and 2 suggest that the performance gains of DyGMamba compared to DyGFormer are marginal on most datasets.

***W2*** Unclear motivation. The motivation of the paper centers on addressing efficiency issues, specifically that existing transformer-based methods struggle to model long histories effectively. The authors propose using the existing Mamba model to tackle this problem. However, the paper lacks statistical information, such as the average or maximum number of historical neighbors, which would provide a clearer outline. Additionally, there are no examples demonstrating DyGFormer’s limitations under these conditions. I also wonder whether it is necessary to model all historical information when dealing with extremely large numbers of historical neighbors (e.g., 8192). In such cases, it may be more effective to consider only the most recent neighbors. Although the authors include an experiment with varying numbers of sampled neighbors, I am curious about the impact of different sampled neighbor counts on performance—specifically, whether increasing the number of neighbors consistently leads to performance improvements and to what extent.

---

> ### Author Response · Authors · 2025-03-11
> **Response to Reviewer TVZU: Part A**
>
> Thank you for the review. Here is our response.
>
> ### Weakness 1
>
> We understand your concern. Let us first re-clarify our contribution:
> 1. Please refer to recent publications that apply Mamba to various domains. For example, Vision Mamba [1] processes image patches as sequences and directly applies Mamba for encoding, similar to the approach used in ViT [2]. These works have garnered significant attention from their respective research communities, and we believe they share a similar level of novelty with our work.
> In our view, adapting Mamba to a new domain—dynamic graph reasoning, in our case—is already a noteworthy contribution. Furthermore, our extensive experiments (Section 4 and Appendix F, G.3, and L) demonstrate that DyGMamba achieves both strong performance and high efficiency simultaneously.
> 2. Additionally, we have introduced a novel Mamba-based module that dynamically selects critical information from node interaction histories, specifically designed to enhance CTDG modeling. Our experimental results (Tables 3, 4, and 5) demonstrate the significance of this dynamic information selection module in improving model performance. We believe this represents a valuable contribution to the field of dynamic graph reasoning.
>
> To supplement our analysis, we provide a comparison between DyGMamba, DyGFormer, and a modified version of DyGFormer where Transformer is replaced with Mamba (referred to as Variant G). Our results (Table 24, Appendix M of our revision) show a performance drop for Variant G across all datasets compared to DyGFormer. Additionally, Variant G performs significantly worse than DyGMamba, with particularly notable declines on the synthetic datasets S1, S2, and S3.
> We attribute this to two reasons:
> 1. Intrinsic Limitations of Mamba: Previous study [3] has shown that Mamba is generally less effective than Transformer in tasks requiring strong copying or in-context learning abilities. In CTDG modeling, strong copying ability is crucial, as many predictions are based on recalling repeated edges. Consequently, replacing Transformer with Mamba in DyGFormer inevitably leads to a performance drop, despite the gains in efficiency.
>
> 2. The Importance of Temporal Pattern Modeling: Our dynamic information selection module plays a crucial role in enabling a Mamba-based model to outperform a Transformer-based model in CTDG modeling, especially when clear temporal patterns exist. This suggests that effectively capturing temporal patterns can significantly enhance the competitiveness of Mamba-based models in this context, further validating the novelty and the contribution of our work.
>
> [1] Zhu, Lianghui, et al. "Vision Mamba: Efficient Visual Representation Learning with Bidirectional State Space Model." International Conference on Machine Learning. PMLR, 2024.
>
> [2] Dosovitskiy, Alexey, et al. "An Image is Worth 16x16 Words: Transformers for Image Recognition at Scale." International Conference on Learning Representations. 2020.
>
> [3] Waleffe, Roger, et al. "An empirical study of mamba-based language models." arXiv preprint arXiv:2406.07887 (2024).

---

> > ### Author Response · Authors · 2025-03-11
> > **Response to Reviewer TVZU: Part B**
> >
> > ### Weakness 2
> >
> > > However, the paper lacks statistical information, such as the average or maximum number of historical neighbors, which would provide a clearer outline.
> >
> > We have added Table 22 in Appendix M presenting the available number of historical neighbors for all datasets in both link prediction tasks under three NSS settings, including maximum and average numbers.
> >
> > > Additionally, there are no examples demonstrating DyGFormer’s limitations under these conditions.
> >
> > The limitations of DyGFormer can be understood from three aspects.
> >
> > 1. High Computational Complexity: As discussed throughout the paper, the Transformer architecture introduces high computational complexity, making DyGFormer inefficient for modeling long histories with limited resources. For instance, as shown in Figure 2 and Table 15, DyGFormer requires 7.57 GB of GPU memory and 47 minutes per epoch when training on the large LastFM dataset, whereas DyGMamba requires only 4.17 GB of GPU memory and 28.45 minutes per epoch. Additionally, Table 16 shows that DyGFormer completes training on LastFM in 2331.2 minutes, while DyGMamba achieves this in just 335.71 minutes.
> > 2. Inability to Capture Temporal Patterns: Table 5 demonstrates that DyGFormer struggles to capture temporal patterns, which we have shown to be crucial for effective CTDG modeling. This issue is particularly evident in datasets with clear temporal patterns, such as S1, S2, and S3. As we have discussed in our response to Weakness 1, while Mamba is generally considered less effective than Transformer in standard settings, our dynamic information selection module significantly mitigates this limitation. In fact, our experimental results consistently show that DyGMamba outperforms DyGFormer, reinforcing the importance of temporal pattern modeling and highlighting a major limitation of DyGFormer.
> > 3. Failure with Limited Historical Context: By examining DyGFormer's performance in Tables 1 and 2 alongside the availability of historical neighbors detailed in Table 22, we observe that DyGFormer struggles on certain datasets (e.g., Wikipedia and UCI) under historical and inductive NSS settings, especially when nodes in the link prediction query have limited one-hop temporal neighbors. This limitation arises because DyGFormer recurrently encodes sequences of one-hop temporal neighbors, and insufficient historical context may cause negative edges to be incorrectly classified as positive, leading to a higher false positive rate. DyGMamba employs a similar encoding approach but benefits from the dynamic information selection module, which better captures temporal patterns, resulting in more stable and improved performance compared to DyGFormer. For example, in transductive/inductive link prediction tasks on Wikipedia under the inductive NSS setting, DyGFormer achieves 62.00±14.00/57.90±11.05, whereas DyGMamba significantly improves performance to 79.86±2.18/71.14±2.44.

---

> ### Author Response · Authors · 2025-03-11
> **Response to Reviewer TVZU: Part C**
>
> > I also wonder whether it is necessary to model all historical information when dealing with extremely large numbers of historical neighbors (e.g., 8192). In such cases, it may be more effective to consider only the most recent neighbors. Although the authors include an experiment with varying numbers of sampled neighbors, I am curious about the impact of different sampled neighbor counts on performance—specifically, whether increasing the number of neighbors consistently leads to performance improvements and to what extent.
>
> This is a valid concern. While we agree that modeling extremely large numbers of historical neighbors is not always necessary, we believe it is still crucial to develop CTDG models capable of efficient reasoning over very long histories.
> We quantify the availability of historical neighbors across various datasets for transductive and inductive link prediction tasks under random, historical, and inductive NSS settings, as shown in Table 22 of our revision. Observations indicate that real-world CTDG datasets, such as LastFM and Enron, typically possess extensive one-hop historical information. Although it is feasible to perform predictions based solely on recent interactions, we posit that effectively leveraging longer historical contexts can further enhance model performance.
>
> From Figure 4, we observe that when considering only 2048 temporal neighbors, DyGMamba consumes 10.85 GB of GPU memory, whereas DyGFormer requires 42.3 GB. While 2048 may not seem excessively large compared to 8192, it remains a significant number in the context of dynamic graph reasoning. In fact, for the DTDG dataset Can. Parl, both DyGFormer and DyGMamba require 2048 temporal neighbors for optimal modeling. As reported in Tables 18–21, DyGMamba substantially outperforms DyGFormer on Can. Parl with 2048 sampled neighbors, particularly in inductive link prediction. Notably, our model achieves nearly perfect predictions under certain settings—for example, 99.57 AP and 99.69 AUC-ROC in transductive link prediction under the random NSS setting. This demonstrates the importance of developing models that can efficiently handle long historical dependencies. Moreover, if future benchmark datasets require even longer historical contexts for effective modeling, DyGMamba would be an excellent fit. The decision to increase the number of sampled neighbors to 8192 in Figure 4 was intended to test the limits of both DyGMamba and DyGFormer using an Nvidia A40 GPU. However, in real-world scenarios where computational resources are more constrained—for example, with a 16 GB GPU—DyGMamba can still effectively model at least 2048 temporal neighbors, whereas DyGFormer would be infeasible.
>
> As you have mentioned, we have already included an experiment with a varying number of temporal neighbors in Appendix G.3. From the result plot (Figure 7 in revision), we observe that, at least for datasets with long-range temporal dependencies (such as Enron), an increase in sampled neighbors positively impacts model performance, particularly for DyGMamba and DyGFormer. However, we also find that for other baseline models, increasing the number of sampled neighbors does not consistently lead to performance improvements. This suggests that the key issue is not whether increasing sampled neighbors is always beneficial for dynamic graph modeling, but rather whether a model is robust enough to effectively process larger amounts of temporal information. To further support this, we conducted additional experiments on Can. Parl using DyGMamba with an increasing number of sampled temporal neighbors (64, 128, 256, 512, 1024 and 2048). The results, presented in the following table (as well as in Table 23 of revision), confirm that DyGMamba consistently benefits from an increasing neighbor count until reaching the large number of 2048, further demonstrating its ability to leverage long-range temporal dependencies.
>
> | Can. Parl    | 64 | 128 | 256 | 512 | 1024 | 2048 |
> | -------- | ------- | ------- | ------- | ------- | ------- | ------- |
> | DyGMamba  | 72.58   | 74.46   | 76.85  | 82.67  | 94.88   | 99.57   |
>
> We have added this discussion into our revision Appendix M. We sincerely appreciate your suggestion and hope our response adequately addresses your concern.

---

> > ### Author Response · Authors · 2025-03-11
> > **Response to Reviewer TVZU: Part D**
> >
> > ### Requested Changes
> >
> > We hope our response to Weakness 2 has provided stronger support for our motivation. We believe this discussion naturally extends from our existing Introduction section. In our revision, we have made only minor adjustments to the introduction to maintain the original flow while incorporating a reference to the complete motivation discussion, which is now included in Appendix M of our revision. Additionally, we have also put the experiment of Variant G in the same section for completeness.
> >
> > Please let us know if this revision meets your requirements. If not, we are happy to make further adjustments as needed.

---

> > > ### Comment · Reviewer_TVZU · 2025-04-08
> > >
> > > Thanks for the response, as well as the additional experiments and clarifications.
> > >
> > > The authors addressed most of my concerns. I highly recommend adding these discussions to the final paper.

---

> > > > ### Author Response · Authors · 2025-04-09
> > > > **Thank You**
> > > >
> > > > Thank you for the response. We will definitely put these discussions to the final paper. Actually, our submitted revision has already included these contents. We might reorganize the order of these additional contents in the final version.

---

### Review · Reviewer_NbKH · 2025-02-25

**Summary Of Contributions:**

This paper proposes a model, DyGMamba, for temporal graph learning. DyGMamba uses state-space-models (SSMs), to encode long temporal dependencies on node and edge embeddings. The main set of experiments are conducted on 7 benchmark link prediction datasets and compared against 9 state-of-the-art methods. The experiments show the proposed model provides superior performance to existing work  and the individual modules of the model are shown to be contributive by ablation studies.

**Audience:**

Yes

**Claims And Evidence:**

Yes

**Requested Changes:**

- A discussion on which settings DyGMamba is expected to perform better than SOTA.
- A discussion on which settings DyGMamba would fail against SOTA.

Please see Strengths and Weaknesses.

**Strengths And Weaknesses:**

- The paper provides a comprehensive list of comparative experiments, ablation studies and efficiency analysis.
- By incorporating longer histories, the study aims to better learn patterns on the graph that are not specific to pairs of nodes, but to lifespan of interactions, e.g. repeated edges. The example on user-song graph provided in the introduction makes sense intuitively. Having said that, I could not locate any experiments or discussion of results that connect back to that intuition. I think you can relate the performance improvements to density of repeated edges. Following paper makes a good analysis of repeating edges on temporal graphs:

    Razieh Shirzadkhani, Shenyang Huang, Elahe Kooshafar, Reihaneh Rabbany, and Farimah Poursafaei. 2024. Temporal Graph Analysis with TGX. In Proceedings of the 17th ACM International Conference on Web Search and Data Mining (WSDM '24). Association for Computing Machinery, New York, NY, USA, 1086–1089. https://doi.org/10.1145/3616855.3635694

- The usage of SSM on the feature matrices built by one-hop neighbourhoods on a temporal setting is an innovative alternative to multi-layer message passing. The order of interactions  provide a natural sequence, thus the potential benefit of SSM on temporal settings is somehow convincing to me. However, there may be a loss of information caused by not considering higher order neighbourhoods in some settings (on datasets that are less vulnerable to over-smoothing). I would love to see a discussion and/or an experiment to address this limitation.
- As also shown by ablation studies, information selection on temporal patterns rather than node level patterns is an important contribution.

---

> ### Author Response · Authors · 2025-03-11
> **Response to Reviewer NbKH: Part A**
>
> Thank you for spending time reviewing our submission. Here is our response.
>
> ### Weakness
>
> > The example on user-song graph provided in the introduction makes sense intuitively. Having said that, I could not locate any experiments or discussion of results that connect back to that intuition. I think you can relate the performance improvements to density of repeated edges.
>
> Thank you for your suggestion. In fact, we have already conducted a related experiment in Section 4.2.3. Specifically, we constructed three synthetic datasets—S1, S2, and S3—each exhibiting distinct temporal patterns. In S1, the density of repeated edges decreases over time as the time intervals between them increase. In contrast, S2 shows an increasing density of repeated edges over time. Meanwhile, S3 follows a periodic pattern.
>
> The evolution of these densities highlights different temporal dynamics. As shown in Table 5, comparing the performance of DyGFormer, Variant A (DyGMamba without the time-level SSM), and DyGMamba reveals that DyGMamba is highly effective in scenarios with clear temporal patterns, as it explicitly captures them through the time-level SSM.
>
> To supplement our analysis, we present another study directly related to the user-song graph example. In this example, we previously noted that a user’s frequency of listening to a song gradually decreases over time. This aligns with the temporal pattern observed in S1, where the density of repeated edges declines over time. Ideally, the model should recognize this pattern and avoid consistently predicting an existing link between every pair of nodes. To evaluate this, we analyze the false positive rate (FPR) produced by CTDG models. FPR is defined as: $\frac{FP}{FP + TN}$, where  FP  (false positives) represents the number of non-existent links incorrectly predicted as existing, and  TN  (true negatives) represents the number of non-existent links correctly identified as non-existing. We report the FPR of Variant A and DyGMamba in the following table.
>
> | Variant A   | DyGMamba |
> | -------- | ------- |
> | 55.23  | 35.11    |
>
> We find that on S1, DyGMamba exhibits a significantly lower FPR compared to Variant A, indicating that modeling temporal patterns with the time-level SSM effectively reduces false positives. This, in turn, prevents the erroneous prediction of a link between the user and the song given an decreasing frequency.
>
> > Following paper makes a good analysis of repeating edges on temporal graphs:
> Razieh Shirzadkhani, Shenyang Huang, Elahe Kooshafar, Reihaneh Rabbany, and Farimah Poursafaei. 2024. Temporal Graph Analysis with TGX. In Proceedings of the 17th ACM International Conference on Web Search and Data Mining (WSDM '24). Association for Computing Machinery, New York, NY, USA, 1086–1089. https://doi.org/10.1145/3616855.3635694
>
> Thank you very much for this reference. We have put it into our citation in Introduction.
>
> > However, there may be a loss of information caused by not considering higher order neighbourhoods in some settings (on datasets that are less vulnerable to over-smoothing). I would love to see a discussion and/or an experiment to address this limitation.
>
> This is an excellent suggestion. We provide the following discussion to address your concern.
>
> 1. Our primary motivation is to design an efficient model that effectively captures long-range temporal dependencies in CTDGs. While incorporating higher-order information is an intuitive approach, it introduces several challenges. Maintaining the same computational budget would require reducing the number of first-order temporal neighbors, leading to a less representative one-hop neighborhood. The alternative—allocating more computational resources such as increased GPU memory—goes against our objective of efficiency in this work. Moreover, modeling higher-order temporal neighbors requires additional specialized modules to differentiate the influence of neighbors from different hops, which can further increase computational cost and difficulties in modeling.
> 2. Recent CTDG models that only consider one-hop temporal neighbors, such as DyGFormer and FreeDyG, have demonstrated strong performance, and DyGMamba further validates this observation. Some of our baselines, such as CAWN and CTAN, do incorporate higher-order information. However, as shown in Table 1, 2, 12, and 13, they are outperformed by DyGMamba in most cases. A potential reason for this is that these models may still struggle to optimally differentiate information from different hops. We acknowledge that efficiently integrating higher-order information into CTDG modeling is a promising research direction, and we would be happy to explore it in future work.
>
> To promote the completeness of our paper, we have put this discussion in Section 4.4 (paragraph "Why We Only Consider One-Hop Neighbors") of our revision.

---

> > ### Author Response · Authors · 2025-03-11
> > **Response to Reviewer NbKH: Part B**
> >
> > ### Requested Changes
> >
> > > A discussion on which settings DyGMamba is expected to perform better than SOTA.
> >
> > DyGMamba is expected to outperform previous state-of-the-art models in the following scenarios: (1) when modeling long-range temporal information is critical; (2) when there is a clear temporal pattern, i.e., the density of repeated edges follows a specific trend. For the first scenario, we have demonstrated in Table 1 and 2 that DyGMamba surpasses previous models on long-range temporal-dependent datasets (LastFM, MOOC, and Enron) in both transductive and inductive link prediction under the random NSS setting. Under the historical and inductive NSS settings, DyGMamba consistently ranks among the top 3 models and even achieves the top 1 position in 7 out of 9 cases. Regarding the second scenario, Table 5 confirms that DyGMamba’s dynamic information selection module significantly enhances its capability to model temporal patterns, enabling superior performance on datasets characterized by clear temporal trends.
> >
> > We have formulated this discussion into the first part of Section 4.4, paragraph "Where DyGMamba Excels and Where It Falls Short" of our revision.
> >
> > > A discussion on which settings DyGMamba would fail against SOTA.
> >
> > DyGMamba might underperform previous state-of-the-art models on certain datasets (e.g., Wikipedia and UCI) under historical and inductive NSS settings, particularly when the number of available one-hop temporal neighbors for nodes in the link prediction query is limited (the availability of historical neighbors across datasets for both link prediction tasks under all NSS settings is presented in Table 22 of revision).  DyGMamba rely on abundant historical information to effectively capture graph dynamics and temporal patterns. Insufficient historical context may hinder its performance, leading to potential misclassification of negative edges and higher false positive rates.
> >
> > We have formulated this discussion into the second part of Section 4.4, paragraph "Where DyGMamba Excels and Where It Falls Short" of our revision.

---

### Review · Reviewer_JfoB · 2025-03-03

**Summary Of Contributions:**

The authors propose DyGMamba for Continuous Time Dynamic Graphs (CTDG), leveraging the recent Mamba model and adjusting it for representation learning on temporal dynamic graphs, which is applied to the task of temporal link prediction. Specifically, DyGMamba is based on a node-level Mamba State-Space Model SSM that encodes the sequence of historical node interactions and a time-level Mamba SSM that extracts edge-specific temporal patterns for the dynamic selection of temporal history interactions. DyGMamba is very competitive with the state-of-the-art DyGFormer method and, for several datasets, outcompetes several CTDG baselines.

**Audience:**

Yes

**Claims And Evidence:**

Yes

**Requested Changes:**

Based on weakness, the following changes could better support the main claims and findings of the paper:
- **[Based on W2]** Authors can expand the discussion on graph-based Mamba by including details and differences with methods designed for static graphs. These could better support the methodological novelty of the proposed approach.
- **[Based on W3]** A justified breakdown of DyGMamba’s preprocessing modules (e.g., features, temporal encoding, patching) and their performance effect in ablations is an important aspect to be covered.
- **[Based on W4]** The efficiency comparison (time, memory) in Fig. 2 can be extended to include larger datasets such as Social Evo. and Reddit, and possibly settings (except for random NSS/transductive) or methods (e.g., CTAN).
- **[Others]** CTDG models events as individual occurrences with precise timestamps, whereas the evaluated datasets are largely discrete. It remains unclear whether DyGMamba can adapt to irregular timestamps. If so, can the authors clarify which model components ensure temporal continuity?

**Strengths And Weaknesses:**

The **main strengths** of this paper can be summarized as follows:
- **(S1)** The authors propose a novel method for CDG built upon Mamba that has recently gathered significant attention, applying it to the challenging task of temporal dynamic link prediction.
- **(S2)** The evaluation for the task of temporal dynamic link prediction is thorough, incorporating comparisons on several datasets and baselines.
- **(S3)** In most cases, DyGMamba offers slightly superior performance compared to CTAN and DyGFormer, while it benefits from improved computation times compared to DyGFormer in the transductive setting. The performance gains of DyGMamba are particularly notable in the transductive/inductive setting and NSS random.
- **(S4)** On the Enron dataset, DyGMamba performs significantly better than DyGFormer when increasing sequence length (i.e., reducing patch count) with fewer parameters and lower training time.

The **weaknesses** of the study are the following:
- **(W1)** In historical and inductive NSS settings, the model is in several cases outperformed or matched in performance by CTAN (and in fewer cases matched by DyGFormer). The efficiency comparison of CTAN in Fig. 2 gives mixed results, but in some cases, CTAN is very competitive.
- **(W2)** The discussion on graph-based Mamba is limited. Discussion in related work could be extended by comparing the model's main blocks to related models, even if they focus on static rather than dynamic tasks, given their shared characteristics.
- **(W3)** The model has complex input processing blocks, e.g., incorporating neighbor features with temporal encoding (from TGAT) and historical interaction frequencies (similar to DyGFormer) while employing neighbor patching to reduce resource usage. However, the necessity of these design choices is not well-justified, and their effect has not been studied in ablations.
- **(W4)** DyGMamba memory- and computation efficiency are improved primarily compared to DyGFormer, which is notably resource-intensive despite being its closest performance competitor. However, several models seem significantly lighter and faster (e.g., the performance-competitive CTAN) than DyGMamba.
- **(W5)** The efficiency comparison in Fig. 2 (time, memory) would be more informative if extended to larger datasets such as Social Evolution and Reddit, similar to how sequence length impact is analyzed in Fig. 3.

---

> ### Author Response · Authors · 2025-03-11
> **Response to Reviewer JfoB: Part A**
>
> Thank you for spending effort reviewing our paper. Here is our response.
>
> ### Weakness 1
>
> We further explain CTAN's behavior from two aspects:
>
> Performance:
> We acknowledge that DyGMamba does not achieve the best performance in every individual case. However, it consistently attains the best overall rank across all datasets and settings. The performance of each CTDG model varies depending on the dataset. Even if we exclude DyGMamba from the comparison and focus solely on baseline methods, none of them consistently achieves the best performance across all settings.
> Regarding CTAN, while it outperforms DyGMamba in certain cases, our model surpasses it by a significant margin in most scenarios—for example, across all settings on MOOC. Furthermore, we have systematically analyzed the scenarios where DyGMamba excels and where it may be outperformed by other SOTA models. Please refer to our response to the Requested Changes proposed by Reviewer NbKH for a more detailed discussion.
> We believe that our extensive evaluation across diverse datasets and settings provides a comprehensive understanding of DyGMamba’s strengths and limitations. Expecting any single model to outperform all baselines in every setting is somewhat unrealistic. Please let us know if further clarification is needed.
>
> Efficiency:
> Note that in Figure 2, we present only the per-epoch training time. However, as mentioned in Section 4.3.1, a key issue with CTAN is its difficulty in convergence. As discussed in Appendix G.2, we provide the total training time for CTAN, DyGFormer, and DyGMamba. From Table 16, we can observe that CTAN requires significantly more epochs to converge compared to DyGMamba. For instance, on LastFM, CTAN takes nearly 54 times of epochs than DyGMamba to reach its best performance, resulting in a substantially longer training time. This is also why CTAN could not be trained within the time limit on the largest dataset, Social Evo., and is marked as “Timeout” in the result tables (e.g., Tables 1 and 2). Our goal is to develop an efficient model that balances effectiveness and training feasibility. While CTAN requires minimal computational resources, this comes at the cost of limited capacity, leading to convergence issues and a suboptimal overall ranking in performance. DyGMamba, on the other hand, achieves the best balance between effectiveness and efficiency. We hope this addresses your concern.
>
> ### Weakness 2
>
> Thank you for your suggestion. We have provided a more detailed discussion in the Related Work section (paragraph "State Space Models for Graphs") of our revision. Please check.
>
> ### Weakness 3
>
> Incorporating temporal and historical interaction frequency features has been validated in previous work. To further address your concern, we present two additional ablation studies to demonstrate their effectiveness in DyGMamba. Specifically, we introduce two new model variants: (1) Variant E that removes the temporal feature; (2) Variant F that removes the historical node interaction frequency feature. We present the results in Table 3 and 4 of our revision. The results clearly indicate that both features contribute positively to performance.
>
> Regarding patching, we have provided a detailed discussion in Section 4.3.2. Please let us know if any confusion remains.
>
> ### Weakness 4
>
> Please refer to our response to Weakness 1.
>
> ### Weakness 5
>
> Thank you for your suggestion. In our revision, we have extended the analysis presented in Figure 2 to include the Social Evo. and Reddit datasets. We have put the additional results (Table 15 and Figure 6 of the revision) in Appendix G.1. We have also provided total training time comparison among DyGFormer, CTAN and DyGMamba on Social Evo. and Reddit in Table 16 of Appendix G.2 of our revision. Our findings on these two datasets align with the conclusions drawn in Section 4.3.1.

---

> > ### Author Response · Authors · 2025-03-11
> > **Response to Reviewer JfoB: Part B**
> >
> > ### Requested Changes
> >
> > > Authors can expand the discussion on graph-based Mamba by including details and differences with methods designed for static graphs. These could better support the methodological novelty of the proposed approach.
> >
> > The extended discussion is presented in Related work (paragraph "State Space Models for Graphs") of our revision.
> >
> > > A justified breakdown of DyGMamba’s preprocessing modules (e.g., features, temporal encoding, patching) and their performance effect in ablations is an important aspect to be covered.
> >
> > Please check our new ablations on Variant E and F. Please refer to our response to Weakness 3 for detailed discussion.
> >
> > > The efficiency comparison (time, memory) in Fig. 2 can be extended to include larger datasets such as Social Evo. and Reddit, and possibly settings (except for random NSS/transductive) or methods (e.g., CTAN).
> >
> > We have included new analysis on Social Evo. and Reddit in Appendix G.1 and G.2 (Table 15, 16 and Figure 6 of our revision).
> > Please refer to our response to Weakness 5 for a detailed discussion. One key clarification is that each model is trained once and then evaluated on transductive/inductive link prediction under every NSS setting. The analysis in Figure 2 pertains solely to the training process and is independent of any evaluation setting. This means that, regardless of the evaluation task (transductive/inductive) or NSS setting (random/historical/inductive), the statistics presented in Figure 2 remain unchanged.
> > Furthermore, the relative differences among models in terms of GPU memory usage, per epoch training time, and the number of trainable parameters are determined solely by model architectures and datasets, rather than the specific evaluation settings. Therefore, we do not find it very necessary to provide additional statistics for different evaluation configurations.
> > We hope this clarification resolves any confusion. Please feel free to reach out if further concerns remain.
> >
> > > CTDG models events as individual occurrences with precise timestamps, whereas the evaluated datasets are largely discrete. It remains unclear whether DyGMamba can adapt to irregular timestamps. If so, can the authors clarify which model components ensure temporal continuity?
> >
> > There appears to be a slight misunderstanding here. Each CTDG dataset used in our study is represented as a stream of events, with each event having its own unique timestamp. As already outlined in Appendix A.1 (Table 6), all datasets considered are tracked using Unix timestamps. Consequently, each event is already treated as an individual occurrence associated with a precise timestamp. We actually have also evaluated DyGMamba on 6 DTDG datasets: Flights, Can. Parl, US Legis., UN Trade, UN Vote, and Contact. The experimental results shown in Appendix L (Tables 18–21) demonstrate that DyGMamba also achieves strong overall performance (in terms of average rank) on these DTDGs for both transductive and inductive link prediction tasks.
> >
> > Concerning the module ensuring temporal continuity, we attribute this capability primarily to two components: encoding temporal neighbors as historical interaction sequences, and learning explicit time encoding features. Furthermore, the time-level SSM explicitly models the intervals between temporally adjacent interactions, effectively capturing continuous variations in edge appearances.
> >
> > We hope this clarification resolves any misunderstanding. Please reach out if further questions remain.

---

### Author Response · Authors · 2025-03-11
**Revision Online**

We thank every reviewer for the constructive reviews and suggestions. We have updated our submission with a revision.

We have highlighted all changes in yellow for better readability.

Here are the key changes:
1. Section 1 and Appendix M: We have slightly changed introduction and link it to a thorough discussion of our motivation in Appendix M. In Appendix M, we have also provided Table 22 that contains the number of available historical neighbors for all datasets on both link prediction tasks under all NSS settings. Besides, we have included an additional experiment on Can. Parl using DyGMamba with an increasing number of sampled temporal neighbors (results in Table 23) to show the merit of modeling long histories. Finally, to further validate our novelty and contribution, we have provided a comparison between DyGMamba, DyGFormer, and a modified version of DyGFormer where Transformer is replaced with Mamba (referred to as Variant G) (results in Table 24). [Requested by Reviewer TVZU]
2. Section 2, paragraph "State Space Models for Graphs": We have added detailed discussion of related works regarding using state space models for graph representation learning. [Requested by Reviewer JfoB]
3. Section 4.2.2, Table 3 and 4: We have introduced two more ablation studies (study E and F), aiming to validate the effectiveness of time encoding and node interaction frequency features. [Requested by Reviewer JfoB]
4. Section 4.4, paragraph "Where DyGMamba Excels and Where It Falls Short": We have added discussion about in what cases DyGMamba excels or falls short. [Requested by Reviewer NbKH]
5. Section 4.4, paragraph "Why We Only Consider One-Hop Neighbors": We have explained in details why we only consider one-hop temporal neighbors when we learn from very long-range historical information. [Requested by Reviewer NbKH]
6. Appendix G, Table 15 and 16, Figure 6: We have added efficiency analysis on Reddit and Social Evo.. We have updated the related efficiency statistics in Table 15 and 16. We have also added a corresponding plot Figure 6. [Requested by Reviewer JfoB]

We hope our revision has addressed your concerns. Please feel free to issue further requests if needed.

---

### Decision · Action_Editor_Cf88 · 2025-04-17

**Recommendation:** Accept as is

**Comment:**

The paper addresses an interesting and important task, continuous-time dynamic graph representation learning, and proposes and investigates a promising novel approach. As such, the findings are likely to be of interest to individuals in TMLR's audience.

The main claims of the paper are that:
(1)	The introduced DyGMamba is the first model to use state-space models for continuous-time dynamic graph representation learning;
(2)	DyGMamba is efficient and effectively models long-term temporal dependencies;
(3)	For the conducted experiments on commonly-used datasets, DyGMamba outperforms the baselines for the dynamic link prediction task.

The paper provides experimental evidence to support these claims.

**Audience:**

Some individuals in TMLR's audience would be interested in the findings.

**Claims And Evidence:**

The claims are supported.